# Production of gas-releasing electrolyte-replenishing Ah-scale zinc metal pouch cells with aqueous gel electrolyte

Feifei Wang[1,2,3,4], Jipeng Zhang[1,3], Haotian Lu[1,2,3,4], Hanbing Zhu[1,3], Zihui Chen[1,3], Lu Wang[1,2,3,4], Jinyang Yu[1,3], Conghui You[2], Wenhao Li[5], Jianwei Song[5], Zhe Weng [1,3], Chunpeng Yang [1,3] ✉ & Quan-Hong Yang [1,2,3] ✉

Aqueous zinc batteries are ideal candidates for grid-scale energy storage because of their safety and low-cost aspects. However, the production of large-format aqueous Zn batteries is hindered by electrolyte consumption, hydrogen gas evolution and accumulation, and Zn dendrites growth. To circumvent these issues, here we propose an "open" pouch cell design for large-format production of aqueous Zn batteries, which can release hydrogen gas and allow the refilling of the electrolyte components consumed during cell cycling. The cell uses a gel electrolyte containing crosslinked kappa (k)-carrageenan and chitosan. It bonds water molecules and hinders their side reaction with Zn, preventing electrolyte leakage and fast evaporation. As a proof-of-concept, we report the assembly and testing of a $Zn||Zn_xV_2O_5 \cdot nH_2O$ multi-layer "open" pouch cell using the carrageenan/chitosan gel electrolyte, which delivers an initial discharge capacity of 0.9 Ah and 84% capacity retention after 200 cycles at 200 mA g$^{-1}$, 370 kPa and 25 °C.

Due to their intrinsic safety and low-cost, aqueous zinc (Zn) batteries have great promise for use as grid-scale energy storage[1–4]. However, the Zn anode suffers from problems including Zn dendrite growth and a parasitic hydrogen evolution reaction (HER) at the anode-electrolyte interface, especially in large-format cells (e.g., multi-layer pouch cells), and this compromises the safety of the batteries[5–7]. Because metallic Zn is thermodynamically unstable in an acidic electrolyte, HER inevitably occurs on the Zn-electrolyte interface during Zn deposition, which decreases the Coulombic efficiency (CE) and cycling stability of the batteries[8,9]. The excessive consumption of electrolyte may result in battery failure, and $H_2$ accumulation triggers battery swelling and even explosion[10]. There may also be dendrite growth due to the "tip effect" which produces dead Zn (i.e., Zn metal regions which are electronically disconnected from the current collector), aggravating hydrogen evolution and even causing short circuits, all of which prevent the large-scale application of aqueous Zn batteries[4].

Many efforts have been devoted to addressing these problems, such as using electrolyte additives, highly concentrated electrolytes[11], nonaqueous electrolytes[12], hydrogel electrolytes[13–15], 3D Zn anodes, and anode/electrolyte interface modification;[16–22] however, most studies were conducted in laboratory-based coin cells, which are quite different from the large-format Zn batteries required for large-scale energy storage[3,23]. The coin-cell or even pouch-cell configurations of Zn batteries reported in the literature are generally adapted from Li-ion batteries[24,25] which have a hermetically sealed closed system because they use hazardous and volatile organic electrolytes and air-sensitive electrode materials[19,26]. However, using sealed Zn batteries not only neglects the fact that the Zn metal and aqueous electrolytes

[1]Nanoyang Group, Tianjin Key Laboratory of Advanced Carbon and Electrochemical Energy Storage, School of Chemical Engineering and Technology, National Industry-Education Integration Platform of Energy Storage, and Collaborative Innovation Center of Chemical Science and Engineering, Tianjin University, Tianjin 300072, China. [2]Joint School of National University of Singapore and Tianjin University, International Campus of Tianjin University, Fuzhou 350207, China. [3]Haihe Laboratory of Sustainable Chemical Transformations, Tianjin 300192, China. [4]Department of Chemistry, National University of Singapore, Singapore 117543, Singapore. [5]State Key Laboratory for Strength and Vibration of Mechanical Structures, Xi'an Jiaotong University, Xi'an 710049, China. ✉e-mail: cpyang@tju.edu.cn; qhyangcn@tju.edu.cn

are air-stable and environmentally-friendly but also imposes avoidable problems, such as $H_2$ accumulation and battery swelling (Fig. 1a and Supplementary Fig. 1)[18,24,27–29]. We therefore propose an open system to resupply any consumed electrolyte component, taking advantage of the safe and abundant aqueous electrolyte, while avoiding the problems of gas accumulation and swelling, to improve the scale and life of the battery. However, other issues such as electrolyte leakage and evaporation could be the problem of the open-system configuration[30–32]. Hydrogel electrolytes show potential in addressing these issues; however, most of the related studies are still limited to small cell size and capacity, not practical for application (Supplementary Table 1)[33–35].

Here, we report a refillable configuration for practical large-format aqueous Zn batteries, which is functioned with an open system for $H_2$ releasing and water refilling and a water-bonding gel electrolyte (Fig. 1b). The gel electrolyte is made of crosslinked biomass-derived kappa (k)-carrageenan and chitosan, which alleviates electrolyte consumption, prevents battery swelling, and mitigates Zn dendrite growth. The crosslinked k-carrageenan and chitosan (CarraChi) gel electrolyte has numerous polar functional groups (-OH, -NH$_2$, -SO$_4^{2-}$) that bond water molecules to suppress fast electrolyte evaporation and the HER. Moreover, the refillable, open-system battery configuration allows effective electrolyte replenishment and gas release in a multi-layer pouch cell configuration. As a result, using the CarraChi gel in refillable, large-format Zn batteries improves the plating/stripping reversibility, cycling stability, and cell lifespan. The refillable symmetric pouch cells demonstrate a life of ~4000 h at a current density of 10 mA cm$^{-2}$ with an areal capacity of 35 mAh cm$^{-2}$, achieving a cumulative capacity of 1286 Ah. In addition, a Zn || ZnxV$_2$O$_5$·nH$_2$O multi-layer pouch cell using the refillable configuration has a high capacity (0.9 Ah) and good cycling stability, retaining 84% capacity after 200 cycles.

## Results

### Water-bonding gel electrolyte

As a critical component of the refillable Zn battery, the water-bonding CarraChi gel was prepared by crosslinking k-carrageenan and chitosan using a simple mixing-casting method and subsequent immersion in 2 M ZnSO$_4$ (Fig. 2a). The protonated -NH$_3^+$ in chitosan interacts with −SO$_4^{2-}$ in k-carrageenan through electrostatic interactions while the -OH functional groups in CarraChi gel can form hydrogen bonds with H$_2$O to decrease the water activity, which is expected to suppress the HER[36]. The k-carrageenan and chitosan

were respectively obtained from biomass-based carrageen and crustaceans. These raw materials ensure the production of the CarraChi solution (Supplementary Fig. 2) and the manufacture of the CarraChi gel by a cast-dry process (Fig. 2b, c). Our rolling and bending tests also prove the feasibility of using the CarraChi membrane in lamination and winding technology for practical battery manufacture (Supplementary Fig. 3a). Scanning electron microscopy (SEM) shows that the morphology of the as-prepared CarraChi membrane is smooth and dense (Supplementary Fig. 3b) with a thickness of 18 μm (Fig. 2d).

We investigated the crosslinking of k-carrageenan and chitosan by X-ray photoelectron spectroscopy analysis (XPS) (Supplementary Fig. 4a, Supplementary Table 2). The N1$s$ XPS spectrum of dry CarraChi gel contains two peaks at 400.3 and 399.5 eV which are respectively ascribed to N-H and C-N bonds in the chitosan (Supplementary Fig. 4b)[37]. In the S2$p$ spectrum, the peaks at 168.9 and 170.1 eV respectively correspond to the S2$p_{3/2}$ and S2$p_{1/2}$ of the sulfate groups in the k-carrageenan (Supplementary Fig. 4c)[38]. We also studied the chemical bonding of the CarraChi gel, in comparison with k-carrageenan and chitosan, by Fourier-transform infrared (FTIR) spectra (Fig. 2e). In the FTIR spectrum of CarraChi gel, the peaks at 1558 and 1449 cm$^{-1}$ are respectively ascribed to the N-H and C-N stretching vibrations in chitosan[37], and the peak at 1219.4 cm$^{-1}$ is attributed to the sulfate stretching of S = O in k-carrageenan[39]. In addition, a red shift of the -OH/-NH$_2$ peak in CarraChi gel is observed, indicating the crosslinking interaction between chitosan and k-carrageenan.

Due to the abundant hydrogen bonds and electrostatic interactions between the k-carrageenan and chitosan, the CarraChi-ZnSO$_4$ has much higher strain-to-failure (45%) and tensile strength (14.2 MPa) than the commercial glass fiber (GF) separator (6% and 0.3 MPa) and a k-carrageenan-ZnSO$_4$ membrane (32% and 4.2 MPa) (Fig. 2f and Supplementary Fig. 5). Compared to the wet CarraChi gel (CarraChi-H$_2$O), the mechanical strength increased after the Zn$^{2+}$ addition, possibly due to the bonding effect between divalent Zn$^{2+}$ and -SO$_4^{2-}$/-OH functional groups in the hydrogel network[40,41]. Such an improved mechanical property of the CarraChi gel electrolyte is expected to facilitate electrode fabrication and battery assembling[42].

### Zn deposition with the gel electrolyte

The CarraChi gel electrolyte shows good electrochemical properties for aqueous Zn batteries. It has a high ionic conductivity of 5.3 mS cm$^{-1}$

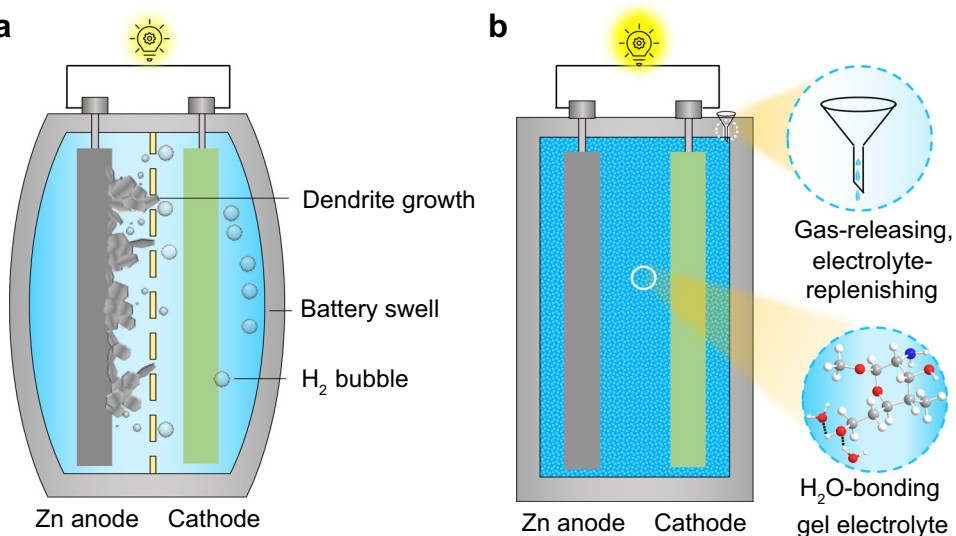

**Fig. 1 | Schematics of Zn batteries operating in two configurations. a** A closed system and **b** an open system. Compared to the closed-system battery with problems such as Zn dendrite formation and H$_2$ evolution, the open-system battery with H$_2$O injection and the gas outlet could improve the cycling life of the cell.

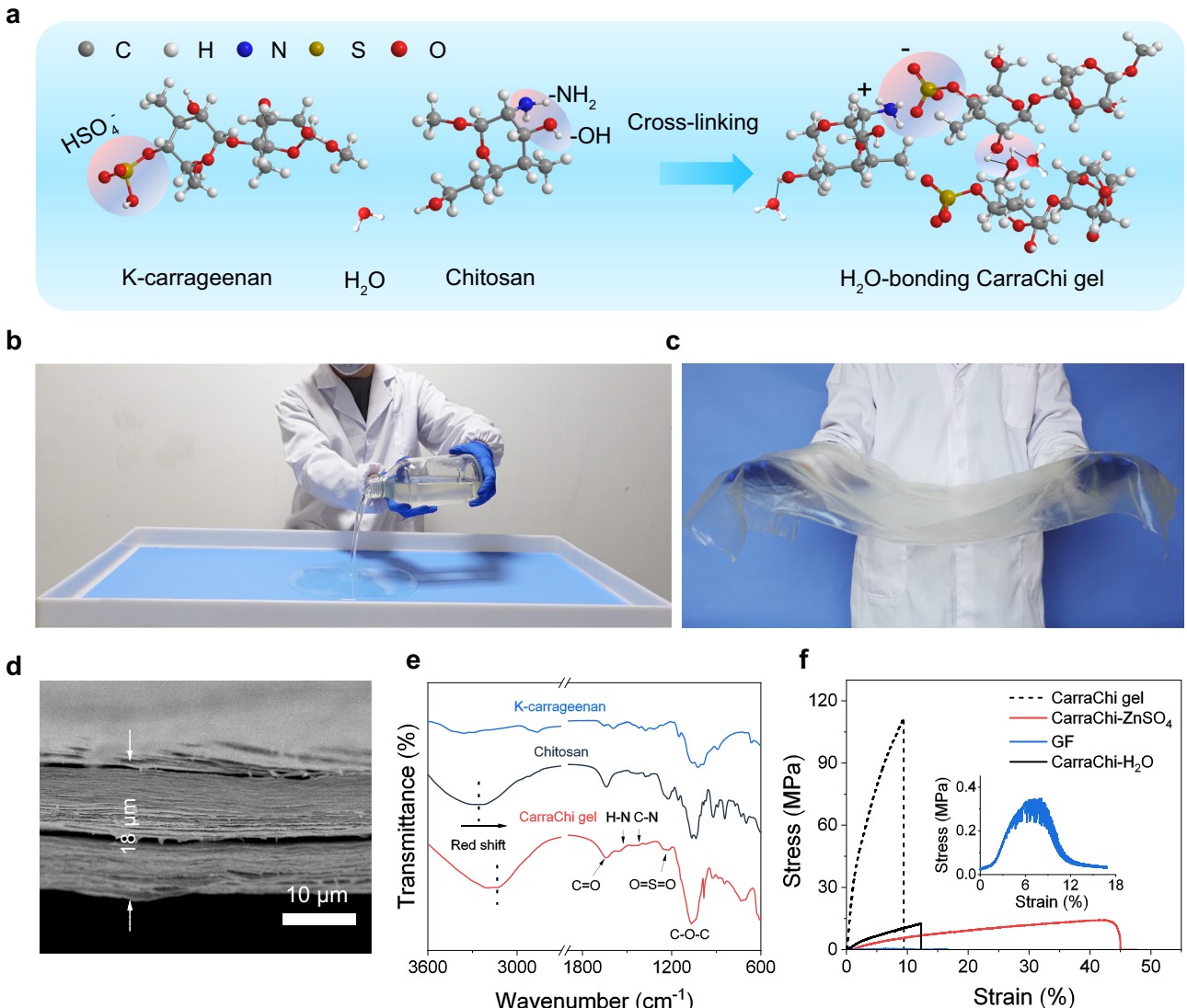

**Fig. 2 | Fabrication and physical properties of the CarraChi gel. a** Crosslinked structure of the CarraChi gel. **b** Photo of the cast-drying fabrication process of the CarraChi gel. **c** Photo of a CarraChi gel membrane. **d** Side-view SEM image of the dry CarraChi gel. **e** FTIR spectra of the dry CarraChi gel, chitosan, and k-carrageenan at 25 °C. **f** Stress−strain curves of the dry CarraChi gel, CarraChi-H₂O, CarraChi soaked with ZnSO₄, and GF at 25 °C.

at 25 °C, which allows $Zn^{2+}$ ion migration through CarraChi at a high current rate. We used linear scanning voltammetry (LSV) to investigate the electrochemical stability window of the CarraChi gel electrolyte (Supplementary Fig. 6) and found that it has a high oxidization potential of more than 2.26 V (vs. $Zn^{2+}/Zn$) compared to that of a $ZnSO_4$ solution (1.99 V vs. $Zn^{2+}/Zn$). The $Zn^{2+}$ transference numbers ($t_{Zn}$) of CarraChi, and GF separator with $ZnSO_4$ electrolyte have also been obtained using the Bruce-Vincent method[43], and the CarraChi gel electrolyte has a high $t_{Zn}$ of 0.52, which exceeds that of the aqueous electrolyte with commercial GF (0.29), indicating the improved $Zn^{2+}$ transfer property in CarraChi gel electrolyte (Supplementary Fig. 7, Supplementary Table 3).

Due to the numerous oxygen-containing functional groups in k-carrageenan and chitosan, the CarraChi gel has a negatively charged surface, as shown by its negative zeta potentials (− 54 mV) (Fig. 3a). After the addition of $ZnSO_4$, the less negative zeta potential of −9.4 mV reveals the effective adsorption or crosslinking of $Zn^{2+}$ by the CarraChi gel, facilitating the desolvation of $Zn^{2+}$ during the deposition process, which differs from GF separator without adsorption toward $Zn^{2+}$ (Supplementary Fig. 8). The desolvation process of $Zn^{2+}$ ion is usually considered to be the rate-determining step for Zn deposition, and its

activation energy ($E_a$) can be obtained using the Arrhenius equation:

$$\frac{1}{R_{ct}} = Ae^{\frac{-E_a}{RT}} \tag{1}$$

where $R_{ct}$ is the charge transfer resistance at the electrode/electrolyte interface, $A$ is the frequency factor, $R$ is the gas constant, and $T$ is the absolute temperature[44,45]. The Zn|CarraChi|Zn cell shows a much smaller $R_{ct}$ than that of the Zn|GF|Zn cell with $ZnSO_4$ electrolyte at temperatures ranging from 25 to 50 °C (Supplementary Fig. 9, Supplementary Tables 4 and 5), confirming the improved charge transfer ability of the CarraChi gel electrolyte. The calculated $E_a$ of the Zn|CarraChi|Zn cell is 27.9 kJ mol⁻¹, which is only 54.5% of that of the Zn|GF|Zn cell (51.2 kJ mol⁻¹) (Fig. 3b), proving easier desolvation in the CarraChi gel electrolyte.

We used Raman spectroscopy to explore the solvation configuration in the CarraChi gel electrolyte (Supplementary Fig. 10). In comparison to the $ZnSO_4$ aqueous solution, the proportion of strong H-bonding water reduces while weak H-bonding water increases in the CarraChi-$ZnSO_4$, suggesting the bonding between the CarraChi gel and $H_2O$ molecules and reconstruction of the H-bond network of free $H_2O$, which are expected to limit HER[29,46,47]. The suppressed HER on the Zn

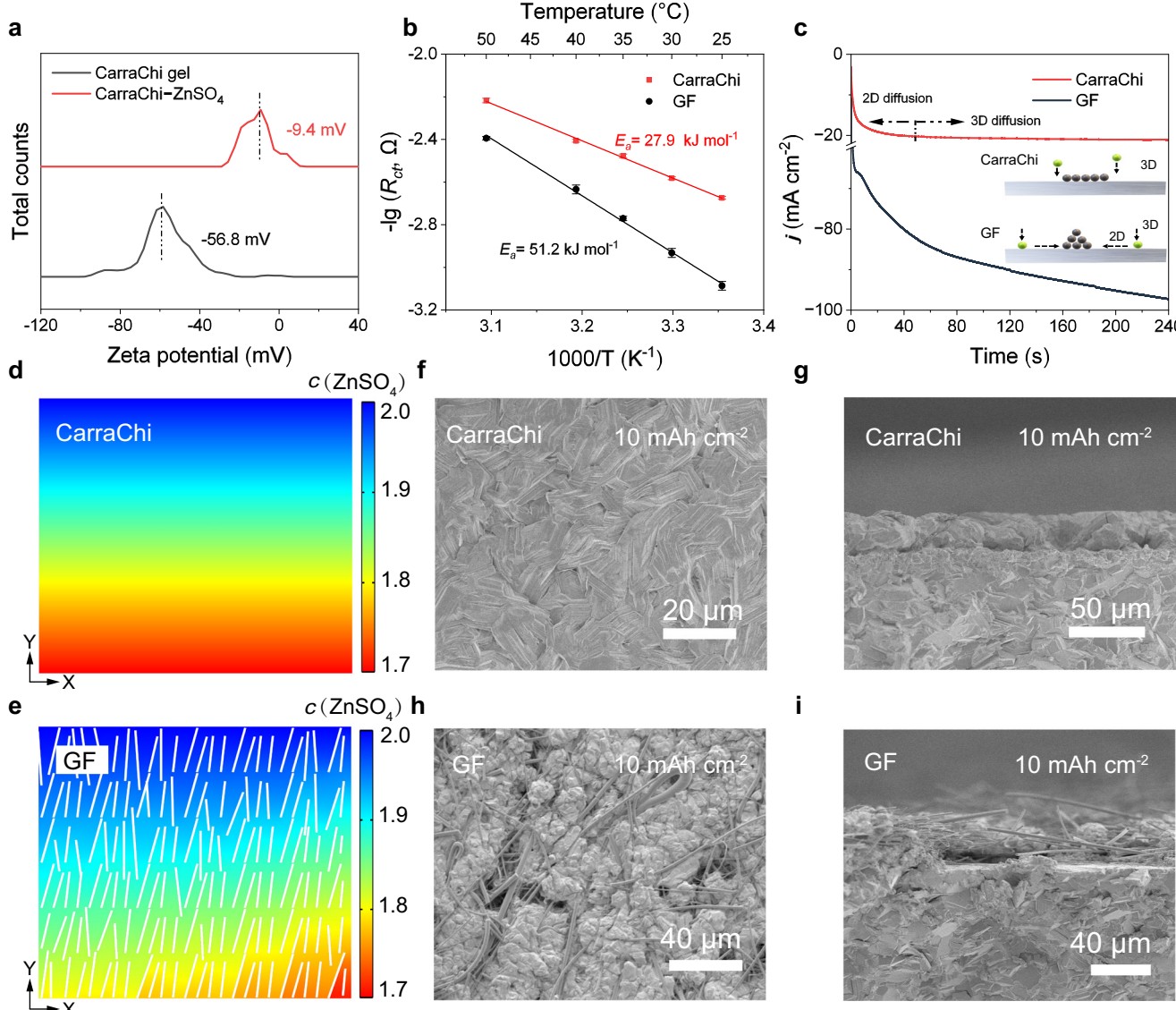

**Fig. 3 | The mechanism of the CarraChi gel electrolyte guiding uniform Zn deposition. a**, Zeta potential of the CarraChi gel electrolyte in an aqueous solution before and after the addition of $ZnSO_4$ (25 °C). **b**, Calculated desolvation energy of $Zn^{2+}$ with a CarraChi gel electrolyte and a GF separator (The error bar represents the error of the fitted $R_{ct}$ value, Supplementary Tables 4 and 5). **c**, Chronoamperometry curves of symmetric devices at a constant potential of −150 mV at 25 °C (inset: schematics of the $Zn^{2+}$ diffusion and reduction on the Zn anode and Zn-CarraChi surfaces). Simulated $Zn^{2+}$ concentration distribution inside **d**, Zn|CarraChi|Zn symmetric cells and **e** Zn|GFA|Zn symmetric cells at 10 mA $cm^{-2}$ (the thin strip represents glass fiber) (*c* represents concentration, M). **f–i** SEM images of deposited Zn in coin cells at 25 °C (10 mA $cm^{-2}$ with deposition capacity of 10 mAh $cm^{-2}$): **f** top and **g**, side views with the CarraChi gel electrolyte; **h** top and **i**, side views with the GF separator.

anode with CarraChi is also proved by a lower onset overpotential (−1.37 V vs. standard hydrogen electrode, SHE) than that with a GF separator (−1.28 V vs. SHE) (Supplementary Fig. 11). In situ optical microscope images further reveal the good suppression of the HER in the competing reactions between Zn deposition and HER on the Zn anode (Supplementary Fig. 12). The CarraChi gel electrolyte also reduces corrosion reactions between the Zn foil and $ZnSO_4$, which is proved by the weaker peak intensity of by-products in the X-ray diffraction (XRD) patterns and the smooth surface in the SEM images in the anti-corrosion experiment (Supplementary Fig. 13).

We also conducted chronoamperometry tests to understand the mechanisms of Zn deposition behaviors at the Zn-CarraChi interfacial sites (Fig. 3c). In the symmetric cell with GF, the response current increased continually beyond 240 s, indicating non-uniform Zn deposition due to the random planar diffusion of Zn ions. In contrast, the response current of Zn-CarraChi stopped increasing within 40 s,

indicating that the surface 2D diffusion process on the Zn surface is suppressed with the CarraChi gel electrolyte[48]. Generally, the absorbed Zn ions tend to diffuse along the surface to energetically favorable sites (tip sites or defective areas) to minimize their energy on the Zn surface. Due to the abundant -OH and $-SO_4^{2-}$ functional groups, the CarraChi gel adhered on Zn foil would provide an extra interfacial energy barrier to prevent 2D $Zn^{2+}$ diffusion. Meantime, the abundant $Zn^{2+}$ transport channels in the crosslinked gel framework render the $Zn^{2+}$ flux uniform. Therefore, a stable and uniform deposition of Zn is formed on the Zn metal anode (Fig. 3c, inset). Using finite-element simulation, we further reveal that the uniform Zn deposition is attributed to the homogenized $Zn^{2+}$ concentration and electric field near the Zn-CarraChi interface (Fig. 3d, e and Supplementary Fig. 14).

Ex situ SEM images show that uniform Zn deposition was obtained with the CarraChi gel electrolyte as the deposition capacity increased from 10 to 30 mAh $cm^{-2}$, which is ascribed to the fast desolvation

kinetics that homogenizes the $Zn^{2+}$ flux and electric field at the electrolyte-electrode interface (Fig. 3f, g, and Supplementary Fig. 15a, b). In contrast, Zn is deposited in the pores of the GF separator with a capacity of 10 mAh cm$^{-2}$ (Fig. 3h, i), followed by the short circuit of the symmetric cell when the capacity increases to 20 mAh cm$^{-2}$, ascribable to the penetration of the Zn metal depositions in the GF separator (Supplementary Fig. 15c, d).

## Long-life Zn battery performance

To test the rate performance and long-term stability of the Zn anode, we performed galvanostatic cycling of symmetric cells using the CarraChi gel electrolyte and the GF separator. The Zn|CarraChi|Zn cell has a good rate performance and stable cycling even at a high current density of 40 mA cm$^{-2}$ compared to the short circuit of the Zn|GF|Zn cell (Fig. 4a). The Zn|CarraChi|Zn cell also shows good cycling stability and long cycling life of over 2500 h at 1 mA cm$^{-2}$ with a capacity of 1 mAh cm$^{-2}$, which is nearly 50 times that of the Zn|GF|Zn cell (Supplementary Fig. 16). Even at a higher current density of 10 mA cm$^{-2}$ with a high areal capacity of 120 mAh cm$^{-2}$, corresponding to a 65% depth of discharge (DOD), a symmetric cell with CarraChi gel electrolyte also shows improved cycling stability with a lifespan of over 180 h (Fig. 4b). Such a high capacity, current density, and DOD are hardly possible for closed-system Zn batteries with GF separators[49]. We used SEM to characterize the changes of the Zn surfaces after 5 cycles at 10 mA cm$^{-2}$ with a capacity of 10 mAh cm$^{-2}$. Compared with the short circuit of the symmetric cell with GF, the Zn anode with the CarraChi gel electrolyte has a smooth Zn surface, indicating uniform Zn plating/stripping during cycling (Fig. 4c). Moreover, in comparison to the unstable Zn plating/striping less than 100 cycles with GF separator, the Zn anode with CarraChi gel electrolyte displays a stable CE for 600 cycles at 5 mA cm$^{-2}$, which is as high as 99.8% after 300 cycles (Fig. 4d, e).

With these advantages of the CarraChi gel electrolyte, we developed a Zn pouch cell with a refillable configuration (Fig. 4f). The size of the Zn electrode was 8 cm × 8 cm, which can be further increased due to the scalability of the CarraChi gel. The pouch symmetric cell has an opening for water refilling and $H_2$ release, which provides a way to supplement the electrolyte and for pressure release. The pouch cell has a long life of ~4000 h (65% DOD) with an areal capacity of 35 mAh cm$^{-2}$ at 10 mA cm$^{-2}$ (Fig. 4g). When the overpotential increased markedly due to electrolyte consumption during cycling, pure water or 2 M $ZnSO_4$ was refiled to sustain the ionic conductivity for normal operation, which is indicated in the voltage-time curves (Fig. 4g). The electrochemical impedance spectroscopy (EIS) curves before and after cycling also confirm that no short circuit occurred during cycling (Fig. 4h). The cells with the refillable configuration also demonstrate an areal capacity per cycle (35 mAh cm$^{-2}$), and a good cumulative capacity (1286 Ah) and average CE (~99.5%), which exceed the Zn||Zn state-of-the-art performance (Fig. 4i and Supplementary Table 6)[38,39,50–57]. In contrast, the Zn|GF|Zn pouch cell exhibits limited capacity, and a short circuit occurs when the current density increases to 2 mAh cm$^{-2}$ (Supplementary Fig. 17).

To explore the practical use of the CarraChi gel electrolyte, we coupled Zn metal anodes with $Zn_xV_2O_5 \cdot nH_2O$ (ZVO) cathodes in coin cell configuration. The ZVO was prepared using a previously published method, and it gave the same SEM images and XRD patterns as reported (Supplementary Fig. 18)[58,59]. The Zn|CarraChi|ZVO cell had high specific capacities of 349.6 mAh g$^{-1}$ at 0.2 A g$^{-1}$ and 200 mAh g$^{-1}$ at 4 A g$^{-1}$ (based on active ZVO mass) in comparison to that of Zn|GF|ZVO cells (Fig. 5a, Supplementary Fig. 19), implying the fast redox kinetics of the Zn|CarraChi|ZVO. As the specific current increased to 8 A g$^{-1}$, a high specific capacity of 115.6 mAh g$^{-1}$ was retained, which is much better than that of Zn|GF|ZVO cells (71.1 mAh g$^{-1}$, 20% capacity retention) (Fig. 5b). In addition, the Zn|CarraChi|ZVO cell had a high specific capacity of 275.8 mAh g$^{-1}$ when the specific current recovered to 1 A g$^{-1}$, indicating its good rate performance.

The improved cycling stability due to effective suppression of dendrite and ZVO dissolution ensures the long life of the Zn batteries (Supplementary Fig. 20). Figure 5c shows the cycling stability of Zn|CarraChi|ZVO and Zn|GF|ZVO cells for 100 cycles at 0.2 A g$^{-1}$. Compared to the fast short circuit and lower CE of the Zn|GF|ZVO cell, the Zn|CarraChi|ZVO cell showed good capacity retention of 88.2% with an average CE of nearly 100%.

We carried out EIS measurements and analyses, and observed the morphology of the Zn||ZVO coin cells before and after cycling. Both Zn||ZVO coin cells had a low ohmic resistance ($R_s$) due to the high ionic conductivity of the aqueous electrolyte (Supplementary Fig. 21a, Supplementary Table 7). Meanwhile, a lower initial charge-transfer resistance was obtained for the Zn|CarraChi|ZVO cell compared to that for the Zn|GF|ZVO cell. Even after cycling, a smaller $R_{ct}$ was measured for the Zn|CarraChi|ZVO cell in comparison to the Zn|GF|ZVO cell (Supplementary Fig. 21b). These results clearly show the faster charge transfer and plating/striping kinetics of Zn with the CarraChi gel electrolyte.

Optical images show that the CarraChi gel retains its initial integrity and transparency (Supplementary Fig. 22a), associated with its high mechanical strength and extensibility. In contrast, the GF became mixed with the cycled Zn due to the "pore-filling effect"[60]. SEM images also show that the cycled Zn electrode with CarraChi had a smooth surface with uniform Zn deposition compared with the rough surface of cycled Zn electrode with GF which contained numerous dendrites (Supplementary Fig. 22b, c).

Considering the high mechanical strength and extensibility of the CarraChi membrane, we also assembled an 8 cm × 8 cm pouch cell by lamination technology and tested its cycling stability (Fig. 5d, e). The Zn||ZVO multi-layer pouch cell with the CarraChi gel electrolyte and open configuration has an initial capacity of 0.9 Ah and good cycling stability, retaining 84% capacity after 200 cycles at 200 mA g$^{-1}$ (based on active ZVO mass), which is better than the Zn||ZVO pouch cells with liquid electrolytes and GF separator (Fig. 5f and Supplementary Fig. 23). This performance is also better than that of the previously reported aqueous Zn metal pouch cells with V-based cathodes (Supplementary Table 8)[61–67]. We ascribe the improved cycling stability to the open, refillable battery configuration, as well as the high strength of the CarraChi, inhibition of side reactions, and dendrite-free Zn plating/stripping caused by the uniform ion flux at the Zn-CarraChi interfaces. The open system allows gas release and electrolyte supplement, and prevents electrolyte depletion and battery swelling.

## Discussion

We have reported an open, refillable battery configuration toward practical large-format aqueous Zn metal batteries, using a water-bonding gel electrolyte composed of k-carrageenan and chitosan, which prevents electrolyte consumption and battery swelling, and achieves Ah-scale capacity Zn batteries for large-scale energy storage. The crosslinked CarraChi gel contains abundant polar functional groups (-OH, -$NH_2$, -$SO_4^{2-}$) to bond $H_2O$ molecules, thus suppressing the fast water evaporation and the HER. As a result, the open and refillable Zn symmetric cells deliver improved plating/stripping capacity, cycling stability, and long life. The open, refillable pouch cells realize a life of >4000 h at a current density of 10 mA cm$^{-2}$ with an areal capacity of 35 mAh cm$^{-2}$ and a DOD of 65%, giving a cumulative capacity of 1286 Ah. A Zn||ZVO multi-layer pouch cell with the CarraChi gel electrolyte and refillable configuration has a high capacity (0.9 Ah) and good cycling stability, retaining 84% capacity after 200 cycles.

## Methods

### Chemicals

Chitosan (D.D ≥ 95%, Aladdin), k-carrageenan (99%, Aladdin), $ZnSO_4$ (ACS purity grade, 99%, Aladdin), $V_2O_5$ (99.95% trace metals basis, Alfa

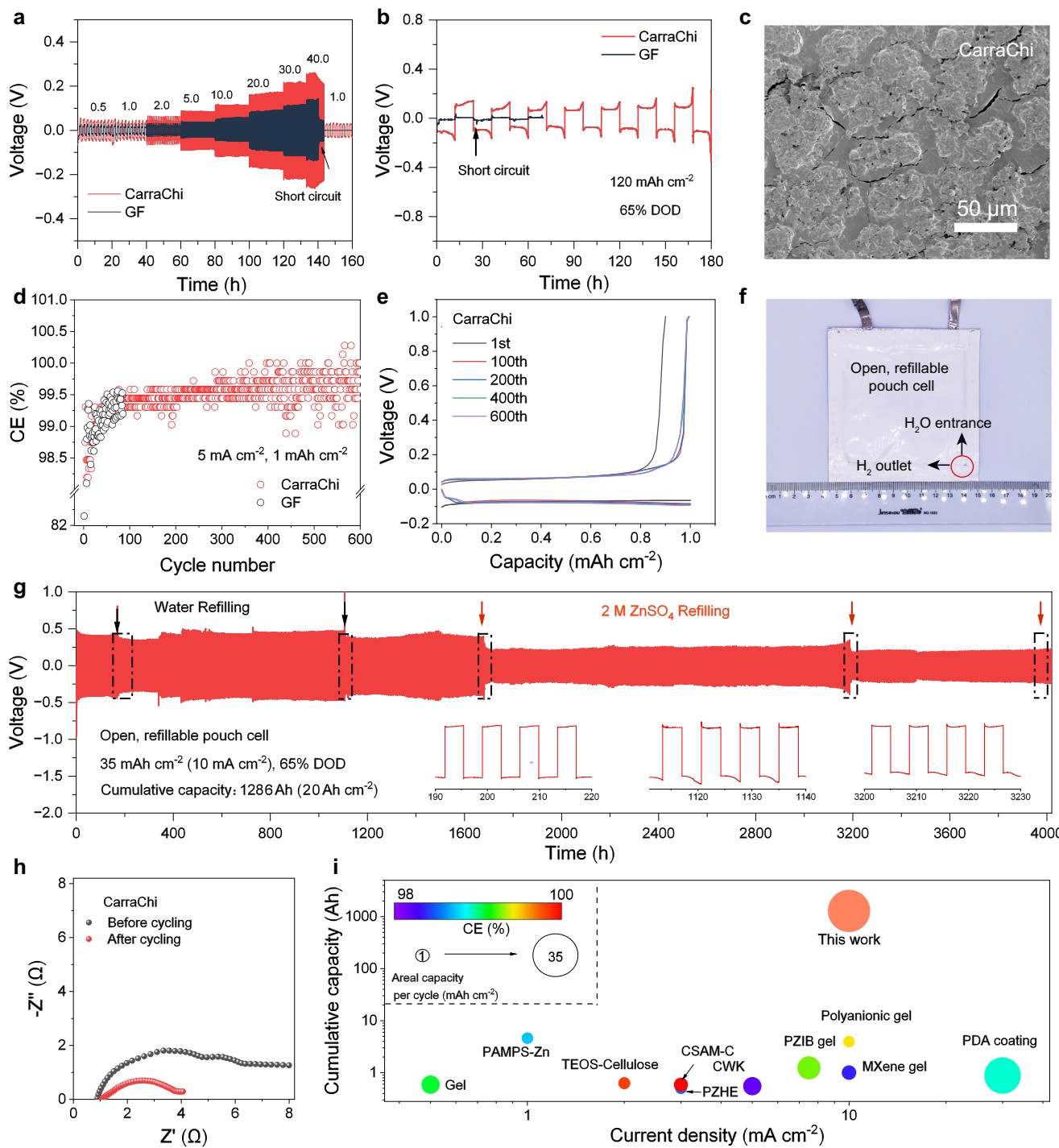

**Fig. 4 | Testing of Zn metal electrode in symmetric coin and refillable pouch cell configurations at 25 °C. a** Rate performance of Zn symmetric cells during Zn stripping/plating at current densities from 0.5 to 40 mA cm⁻² with CarraChi gel electrolyte or the GF separator. **b** Cyclic performance of Zn symmetric cells in coin cells at 10 mA cm⁻² with a plating/stripping capacity of 120 mAh cm⁻². **c** SEM image after 10 cycles at 10 mA cm⁻² (10 mAh cm⁻²) using the CarraChi gel electrolyte. **d** CE values of Zn plating/stripping in Zn|CarraChi|Cu and Zn|GF|Cu cells at 5 mA cm⁻² with a capacity of 1 mAh cm⁻². **e** Galvanostatic voltage profiles of the Zn|CarraChi| Cu cell. **f**, Optical image of the Zn|CarraChi|Zn pouch cell with the open and refillable configuration (8 cm × 8 cm). **g** Cycling performance of pouch Zn symmetric cells at 10 mA cm⁻² with the refillable system (inset: enlarged voltage-time curves). **h** EIS curves of the Zn|CarraChi|Zn symmetric cells before and after cycling (charging state) for 4000 h. **i** Performance comparison of the Zn|CarraChi|Zn pouch cell with other Zn symmetric cells reported in the literature, showing the CE, current density, and cumulative capacity (color represents the CE values).

Aesar), Zinc acetate (Zn(Ac)₂, ACS purity grade, 98%, Aladdin), N-methyl-2-pyrrolidone (NMP, 99%, Aladdin), deionized water (18.2 MΩ cm⁻¹), Super P carbon (99.5%, Guangdong Canrd New Energy Technology Co., Ltd, particle size of 40 nm).

## Preparation of the CarraChi gel electrolyte

1 g chitosan was added to 1000 ml deionized water under magnetic stirring for 1 week at 25 °C. The saturated chitosan solution was obtained by filtrating this solution to remove any undissolved material.

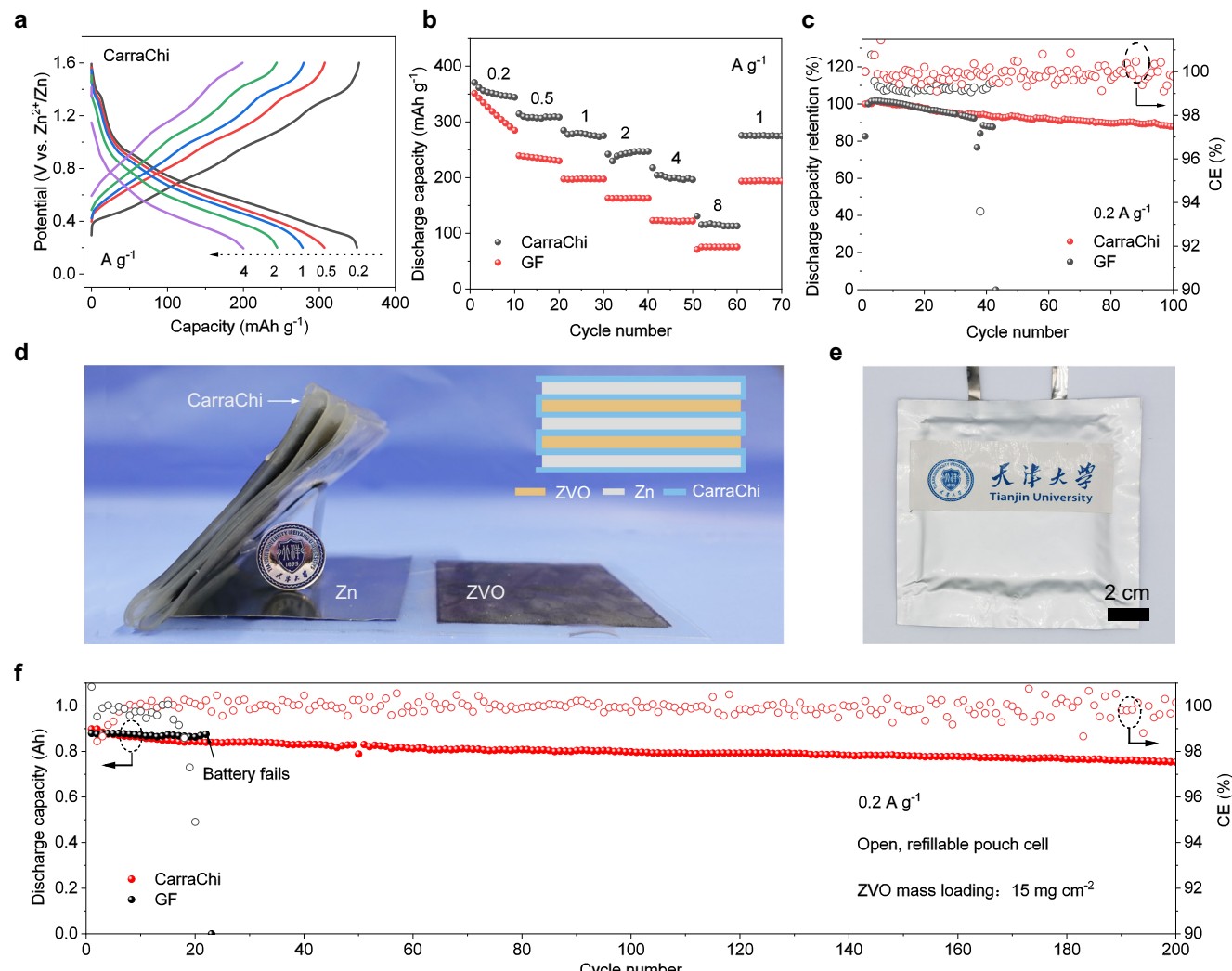

**Fig. 5 | Application of CarraChi gel electrolyte in Zn | |ZVO coin and pouch cell configuration (25 °C).** These measurements are based on the active ZVO mass in coin cells **a**–**c**: **a** Voltage profiles of Zn|CarraChi|ZVO at various charge/discharge specific currents. **b** Comparison of the rate performance of Zn|CarraChi|ZVO and Zn|GF | ZVO batteries at different specific currents from 0.2 to 10 A g⁻¹. **c** Capacity retention and CE curves of the Zn|CarraChi|ZVO battery in a cycling test at 0.2 A g⁻¹ compared to those of a Zn|GF | ZVO battery. **d** Schematic of the lamination process and corresponding side view optical image of the laminated Zn|CarraChi|ZVO. **e** The Zn|CarraChi|ZVO pouch cell. **f** Cycling stability of the pouch cell with CarraChi and GF at 0.2 A g⁻¹ (based on active ZVO mass, 25 °C).

The concentration of the saturated chitosan solution is -0.18 g L⁻¹ at 25 °C. To prepare the CarraChi gel precursor solution, 1 g k-carrageenan was dispersed in 100 ml of the saturated chitosan solution with magnetic stirring for 3 days. 50 ml of the as-prepared solution was cast into a PTFE mold with a diameter of 9 cm and allowed to stand for drying in the air environment. The obtained pure CarraChi was immersed in a 2 M ZnSO₄ solution (12 hours) for use (CarraChi).

### Preparation of the $Zn_xV_2O_5·nH_2O$ cathode

A total of 3 mmol of commercial $V_2O_5$ and 2.4 mmol of $Zn(Ac)_2$ were dissolved in 70 mL deionized water, then 5 mL acetone and 2 mL 10 wt % $HNO_3$ were then added to the solution with wet ultrasonication (70 W) in water for 5 min. The mixture was transferred to a 100 mL Teflon-lined autoclave (Anhui Kemi Machinery Technology Co., Ltd) and heated at 180 °C for 24 h. After natural cooling, the $Zn_xV_2O_5·nH_2O$ powder was obtained after filtration and washing with deionized water, followed by drying in a vacuum at 80 °C for 12 h.

### Material characterization

Scanning electron microscopy (SEM, Hitachi S4800) was used to characterize the morphology and elements. The cycled cells were disassembled under an air environment and the electrodes were taken out and rinsed with ethanol several times to wash away any electrolyte salt and attached glass fiber. The electrodes were air-dried at 25 °C and then carefully transferred into the SEM sample chamber. A cross-sectional SEM photo was taken to measure the deposition thickness of Zn after a similar process. XRD patterns were collected on a Bruker D-8 diffractometer (Cu Kα radiation, λ = 0.154 nm) at 25 °C. The XPS measurements were conducted using an ESCALAB 250Xi (Thermo Fisher) with a monochromatic Al Ka source. Raman spectra were tested by a LabRAM HR spectrometer from Horiba, which was equipped with an argon ion laser operating at a wavelength of 532 nm. The FTIR technique, performed using the Thermo Scientific Nicole iS20 instrument, was utilized to acquire the relevant data concerning the covalent bonds. A mechanical testing machine (SUST CMT-1103) operating at a strain rate of 1.0 mm min⁻¹ was employed to evaluate the mechanical properties of the separators and gel electrolyte. Zeta potential measurements were conducted using the ZETA SIZER Nano series at 25 °C. The GF separator was ground and pulverized into powders for testing zeta potential. The anti-corrosion test included assembling symmetric cells with GF/ZnSO₄ and CarraChi gel electrolytes, respectively, and allowing them to remain undisturbed under open-circuit voltage for

one week. Afterward, the Zn electrodes were extracted, rinsed with deionized water, and air-dried at 25 °C for further characterization.

## Assembly of coin cells

Zn foils (99.999%, Chengshuo) were polished with 1000 mesh sandpaper and directedly used as Zn electrodes. 2032-type coin cells were assembled with two identical Zn foils ($d = 10$ mm, the thickness of 100 µm) after polishing. CarraChi gel or a GF (type A, Whatman, thickness of 0.26 mm, porosity of 90%, average pore of 0.7 µm) was used as the separator. For full cells, the cathodes were made by casting a mixture that contained ZVO powder (70 wt%), Super P carbon (20 wt%), and PVDF (10 wt%), onto a carbon cloth (produced by CeTech, porosity: 77%, thickness: 10 µm, denoted as CC). The CC loaded with ZVO (ZVO/CC), Zn foil, and CarraChi were used as the cathode, anode, and electrolyte, respectively (the mass loading of the active ZVO in the cathode: 1 mg cm$^{-2}$).

## Assembly of pouch cell

The pouch symmetric cell was assembled with two identical Zn foils ($8 \times 8$ cm$^2$) polished with 1000 mesh sandpaper. CarraChi gel, or a GF (GF-A, Whatman) was used as the separator. As for the pouch full cell, the ZVO was coated on carbon cloth ($8 \times 8$ cm$^2$) with Super P and PVDF binders (the ZVO mass loading of 15 mg cm$^{-2}$), and the CarraChi gel electrolyte was cut into custom size for lamination and winding. Concerning the Zn|CarraChi|ZVO pouch cell, a pre-operation injection of 2 ml of 2 M ZnSO$_4$ solution was carried out. As for the Zn|GF | ZVO pouch cell, the previously mentioned gel electrolyte has been replaced with a GF separator, and concurrently, a 5 ml injection of 2 M ZnSO$_4$ is administered. Ti foil ($1 \times 4$ cm$^2$) was connected with the current collector as the tab. After lamination, the assembled battery was encapsulated in an aluminum-plastic film. A pore was punched on the side of the sealing bag away from the tag. The water refilling procedure was conducted when the overpotential increased in comparison to the initial overpotential, and 0.5 ml g$^{-1}$ (ratio of electrolyte to the active ZVO) electrolyte was injected every time. The pouch cell is subjected to a pressure of 370 kPa and a temperature of 25 °C during the electrochemical measurements.

## Electrochemical measurements

Galvanostatic charge and discharge measurements were carried out with a Neware battery test system. The specific currents and discharge capacities in full cells are calculated based on the mass of active ZVO in the cathode. To improve the reproducibility of the experiment, we conducted a minimum of two repetitions for the electrochemical performance. The cycling stability tests were performed at various specific currents. LSV was performed on an Autolab Analyzer PGSTAT 128 N (Metrohm, Switzerland). EIS was measured on an IVIUM electrochemical station. For the HER test, the LSV was tested in a 1.0 M Na$_2$SO$_4$ aqueous solution, with a scan rate of 5 mV s$^{-1}$. The EIS tests were measured in the potentiostatic method, the frequency is in the range from 0.1 Hz to 100 kHz at open circuit potential and an amplitude of 5 mV. The open-circuit voltage time applied before EIS is 10 min, and the data number of EIS is 61. The ionic conductivity of the CarraChi gel was tested by two blocking electrodes (stainless steel, denoted as SS; Tianjin Yide times Technology Development Co., LTD; thickness: 100 µm; diameter: 1 cm) and calculated according to the following equation:

$$\sigma = \frac{L}{RS} \qquad (2)$$

where R is the resistance according to EIS measurement (2.5 Ω), L is the thickness of the CarraChi gel (0.105 mm), and S is the area of the contact between the SS and the gel (0.785 cm$^2$). The ionic conductivity of the CarraChi gel electrolyte was determined to be $\sigma = 5.3$ s mS cm$^{-1}$.

## Finite-element simulation of symmetric cells

A two-dimensional continuum model was used to investigate the ionic and potential distributions in Zn metal symmetric cells (Zn|CarraChi|Zn, Zn|GFA|Zn) at a constant current density ($i_0 = 10$ mA cm$^{-2}$)[68,69]. In CarraChi and ZnSO$_4$/GF electrolytes, there are the mass balance and the charge conservation. The ionic flux density was described by the Nernst-Planck equation with diffusion and migration terms. At the Zn-electrolyte interface, the boundary conditions were set according to the cell test system. The other boundaries were set as no flux. All finite-element simulations were conducted in COMSOL Multiphysics 6.1. The initial conditions and ionic diffusion coefficients were set to be consistent with the electrochemical measurement conditions and results.

## Reporting summary

Further information on research design is available in the Nature Portfolio Reporting Summary linked to this article.

## Data availability

The data that support the findings of this study are available in the online version of this paper and the accompanying Supplementary Information, or available from the corresponding authors on reasonable request. The source data underlying Figs. 2e, f, 3a–c, 4a, b, d, e, g, h, and 5a–c, f are provided as a Source Data file. Source data are provided with this paper.

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

## Acknowledgements

This research was supported by grants from the National Key Research and Development Program of China (No. 2022YFB2404500). We also appreciate the funding supports from the Haihe Laboratory of Sustainable Chemical Transformations, National Industry-Education Integration Platform of Energy Storage, and the Fundamental Research Funds for the Central Universities.

## Author contributions

Q.-H.Y. and C.Y. supervised the project. C.Y. and F.W. conceived the idea. F.W. and J.Z. synthesized the materials and performed the characterizations and electrochemical measurements. H.Z., Z.C., L.W., W.L., J.S., and J.Y., contributed to the structural characterizations and electrochemical measurements. H.L. carried out the finite simulation. F.W., C.Y. Z.W., and Q.-H.Y. organized and wrote the manuscript. All authors contributed to the discussion and revision of the manuscript.

## Competing interests

The authors declare no competing interests.
