## [Peer Review File · Nature Communications]

REVIEWER COMMENTS

Reviewer #1 (Remarks to the Author):

The manuscript studies the critical problem of aqueous zinc-ion batteries (ZIBs). Zinc anode in aqueous electrolytes suffers from corrosion and hydrogen evolution reaction (HER). Using an open-structure cell can prevent hydrogen build-up in the cell while allowing replenishment of electrolytes. The study uses a polymer gel electrolyte, a crosslinked carrageenan with chitosan. The study carried out the tests in pouch cells. Though the proposed cell concept is new and has high potential, the study needs more results and an advanced experimental setup to support the hypotheses and improve the insight into the phenomena that took place in the system.

The abstract is adequate and contains essential elements.

The introduction covers the problem statement. The authors describe several approaches to solve the issues of zinc-anode. The authors must include the use of nonaqueous electrolytes (10.1002/batt.202100361). Because several works have studied polymer gel electrolytes for ZIBs, the authors must highlight the shortcomings and research gaps. Notably, the research hypotheses should be presented explicitly.

The authors refer Zn/CarraChi/Zn battery and Zn/GF/Zn battery. These are not batteries; they are symmetrical cells.

The concept of replenishing the electrolyte is fine. However, it needs to be clarified when and how much to fill the electrolyte. Also, the EIS of the system during cycling should be monitored.

It is unclear how tiny hydrogen bubbles leave the zinc surface, mainly when the viscous gel electrolyte covers the surface. Can some of these tiny hydrogen bubbles possibly accumulate on the surface and isolate the zinc surface from the electrolytes?

It is essential to elaborate on the effects of the gel polymer electrolyte regarding the mitigation of zinc surface passivation by zinc hydroxide sulfate.

The capacity of the cell is measured to the cut-off voltage approaching 0 V, which is not practical for real applications. Anyway, the authors should compare their results with other results reported for liquid electrolytes of vanadium-oxide-based cells.

Recently, several works have proposed different polymer gel electrolytes; the authors must compare the advantages and disadvantages of these works (10.1038/s41598-020-69521-x, 10.1016/j.electacta.2022.141365, 10.1021/acsami.1c05941).

Reviewer #2 (Remarks to the Author):

This manuscript proposed a new gel electrolyte that can minimize the problems in Zn metal electrodes, such as hydrogen evolution reaction (HER) and nonuniform Zn deposition reaction. The authors claimed that the proposed mixed gel & liquid electrolyte (CarraChi gel-ZnSO₄) can bond water molecules and reduce the HER, and possesses high mechanical strength to prevent the Zn dendrite issue. Also, they argued that the high transference number of Zn ions and the facilitated desolvation process of Zn ions in CarraChi gel-ZnSO₄ enhanced the Zn redox reaction. Zn batteries using the CarraChi gel-ZnSO₄ showed surprising performances for both half-cells and full-cells.

It was quite clever that the authors made a hole in the pouch cell to refill the water and to let the gaseous side-reaction products easily be removed away. I also personally believe that there is no reason for aqueous batteries to follow the conventional closed-type configurations of alkali metal-based batteries considering that the aqueous batteries can retain air or humid conditions. So, I do think proposing a new configuration, although there was a small change, can encourage researchers in the relevant field to think more broadly about cell design. However, I see there are lots of ambiguous speculations which require precise explanations, and more detailed comparisons with the reference Zn batteries are necessary. In light of all these circumstances, a rigorous major revision of this paper is recommended. The comments are shown below. Without precise explanations of every comment, this paper should be published in other journals.

Comments

1. First of all, why would the formation of hydrogen bonds between the water molecules and CarraChi gel-ZnSO₄ reduce the HER? I know it is a conventional argument typically used in aqueous battery systems, but I don't see the scientific background of this argument. Also, the part saying that "the decrease of the non H-bond contents indicates a weaker water activity" should be clearly explained. Perhaps, the authors' perspectives on changes in electronic structure, and experimental results on changes in solvation structure might be meaningful.

2. The authors showed LSV data to claim that CarraChi gel-ZnSO₄ can suppress the HER. However, it is questionable why the authors obtained the overpotential at 5 mA cm⁻². Normally, we are interested in the on-set potential of HER. If the authors want to claim the kinetically suppressing effect of CarraChi gel-ZnSO₄ on HER in the way they used in the paper, it is necessary to show that the overpotential of the Zn redox reaction is lower than the reference case. Otherwise, the method the authors used will show the effects of low ionic conductivity (5.3 mS cm⁻¹, which is lower than the typically reported ionic conductivity of aqueous 2M ZnSO₄ electrolyte), which will obviously show a higher overpotential for both the Zn redox reaction and HER. By the way, you should change Supplementary Fig.12 to Supplementary Fig. 10 on page 10, and Supplementary Fig. 10 to 11 on page 12.

3. Where did the C-N-C bond come from in Supplementary Fig. 3b? Also, I believe Fig. 2g needs to be revised. In the manuscript, it is said that Fig. 2g is about the FT-IR result comparing the spectrum of carrageenan and chitosan with CarraChi. However, there are only spectra of CarraChi and CarraChi gel-ZnSO₄ in Fig. 2g. As a result, I could not clearly observe the red shift of the -OH/-NH₂ peak in CarraChi.

4. I am not sure if the part saying the "high mechanical strength of CarraChi gel-ZnSO₄ can suppress the dendritic growth of Zn metal" is really reasonable. It is known that the shear modulus of the surface layer should be at least 2 times higher than that of the metal electrode to mechanically induce uniform metal deposition reaction and prevent penetration.

5. In Supplementary Fig. 6, I think it was too short to observe true steady state current for both Zn|CarraChi|Zn and Zn|GF|Zn. Also, I think it is really confusing that the authors use both the terms CarraChi and CarraChi gel-ZnSO₄ throughout the paper. As far as I understood, it should be CarraChi gel-ZnSO₄. Am I right?

6. Please provide the zeta potential of GF in a neutral solution before and after the addition of ZnSO₄.

7. The authors say a high transference number of Zn ions, and a fast desolvation process for Zn ions in CarraChi gel-ZnSO₄ can result in faster reaction kinetics of Zn metal electrodes compared to the GF case. It seems true since the R_{ct} of Zn/Zn cell using CarraChi gel-ZnSO₄ is lower than that of the cell using GF. However, the overpotential of the Zn/Zn cell using CarraChi gel-ZnSO₄ is much higher than that of the

cell using GF. Is this due to the low ionic conductivity of CarraChi gel-ZnSO₄? Also, the authors claimed that using CarraChi gel-ZnSO₄ can increase the CE of Zn/Cu cells. Due to some error points that show abnormal CEs for CarraChi gel-ZnSO₄ cell, the average CE of cell using CarraChi gel-ZnSO₄ might be similar or a little higher than the cell of GF, but it seems like Zn/Cu cell using GF has higher CEs in most of the cycles before the 150th cycle. Please provide reasonable explanations for these circumstances.

8. Please provide the current density condition used for comparing the SEM image in Fig. 3f, and Supplementary Fig. 9a, b).

9. Why is the rate capability of the full cell using CarraChi gel-ZnSO₄ better than the reference case? I believe using CarraChi gel-ZnSO₄ can impinge the overpotential as is proved in Zn/Zn cell test. Please also provide the voltage profile of the Zn|GF|ZVO and compare it in Fig. 5a.

10. Lastly, please compare the refillable pouch cell test result in Fig. 4g and Fig. 5f with the GF cells.

Reviewer #3 (Remarks to the Author):

The present manuscript proposed a hydrogel that can be prepared on a large scale and a refillable battery design. Indeed, the overall electrochemical performance of the ZIBs has been improved in this study. However, the work fails to reflect the big progress in the development of high-performance ZIBs. Many studies proposed hydrogel systems. The current work just selected a different raw material that has been commercialized. The proposed battery design also disables addressing the key challenge of ZIBs, such as HER and long-cycling stability of the cathode. In all, the proposed concept of the scalable synthesis of hydrogel and battery design can not present an impressive innovation. Thus, I do not support the acceptance of the manuscript for publication in the present form.

More issues are listed as follows:

1. The authors proposed a hydrogel enabling to suppress the electrolyte loss. However, why do the authors need to refill the water? Refilling water means that the electrolyte will be consumed or can not be preserved very well in this system. In this situation, there is an obvious contradiction between the proposed concept and experimental results. Meanwhile, the authors should clarify why the electrolyte is consumed or can not be preserved even though hydrogel has been used in this manuscript. Does the refilled water affect the electrolyte concentration and further the electrochemical performance? If it does, it represents that the as-proposed battery design may fail to maintain the electrochemical

performance. If it doesn't, the authors need to elaborately explain why the electrochemical performance does not change even though the electrolyte concentration changes after refilling the water. Additionally, as claimed that the hydrogel can suppress HER, but why do the authors still need to design a gas outlet?

2. K-carrageenan and chitosan have been successfully commercialized for many years. Therefore, large-scale preparation of hydrogel using these two raw materials is not an innovation of this manuscript.

3. Figure 2h demonstrated that the tensile stress of Carrachi-ZnSO₄ has evidently decreased compared with CarraChi, suggesting that some chemical bonds have changed in this gel because of Zn²⁺. Please make a detailed explanation of what happened to the gel when Zn²⁺ was added. Authors should also supplement the visualization proof to demonstrate the gel can prevent dendrite penetration.

4. Which type of GF is used in this work? The authors should mention the details of GF in the experimental section. This is very important for those who want to repeat this experiment. In addition, the thickness of GF should be at around 50 μm at least. Unfortunately, the thickness of CarraChi gel and GF is different when the authors compare the electrochemical performance, which may affect the conclusions. Please compare the electrochemical performance with the same thickness of CarraChi and GF.

5. Could the author describe what kind of barrier the gel layer provides for Zn surface diffusion?

6. It will be nice if the author could perform some simulations to visualize the electric field at the electrolyte-electrode interface that has been homogenized when gel electrolyte is used in this study.

7. "1.26V vs RHE at 5 mA cm⁻² (S Fig. 12)" Please change Fig. 12 to Fig. 10. In Fig. 10, it indeed verifies HER can not be fully suppressed even using CarraChi gel because the onset potential for HER is almost the same, in which both start around -1.1V. The difference is that the gel system shows a smaller current density compared with GF. In this case, it is incorrect to claim Fig. 10 demonstrates slower evolution kinetics. Importantly, please use ZnSO₄ electrolyte rather than NaSO₄ electrolyte to test LSV.

8. The calculation of average CE in the symmetric system in this manuscript is wrong. Please refer to Nat Energy 5, 743–749 (2020). <https://doi.org/10.1038/s41560-020-0674-x> for your average CE calculation.

9. It should be more subjective when the authors conclude. The good capacity retention of Zn|CarraChi|ZVO is not only related to the effective suppression of Zn dendrite growth but also the inhibition of self-corrosion and side reactions of Zn foil.

10. The author should explain why ZVO in GF almost shows the same specific capacity as CarraChia before 40 cycles in Figure 5c, while the specific capacity of GF is higher or lower than CarraChi in Figure 5b. Please mention which current density has been used in Figure 5f.

Point-by-point response to the comments

Reviewer #1: The manuscript studies the critical problem of aqueous zinc-ion batteries (ZIBs). Zinc anode in aqueous electrolytes suffers from corrosion and hydrogen evolution reaction (HER). Using an open-structure cell can prevent hydrogen build-up in the cell while allowing replenishment of electrolytes. The study uses a polymer gel electrolyte, a crosslinked carrageenan with chitosan. The study carried out the tests in pouch cells. Though the proposed cell concept is new and has high potential, the study needs more results and an advanced experimental setup to support the hypotheses and improve the insight into the phenomena that took place in the system.

Reply: We thank the reviewer for the valuable comments. We have supplemented more experimental setups and results to support our hypotheses based on the reviewer's comments and provided point-by-point responses below.

1. The abstract is adequate and contains essential elements.

Reply: We thank the reviewer for the positive comment.

2. The introduction covers the problem statement. The authors describe several approaches to solve the issues of zinc-anode. The authors must include the use of nonaqueous electrolytes (10.1002/batt.202100361). Because several works have studied polymer gel electrolytes for ZIBs, the authors must highlight the shortcomings and research gaps. Notably, the research hypotheses should be presented explicitly.

Reply: We thank the reviewer for the constructive suggestion. We have added a discussion on using nonaqueous electrolytes for ZIBs citing the recommended paper (10.1002/batt.202100361, Ref. 12), which mainly reviews the development of nonaqueous electrolytes for ZIBs and mechanisms of nonaqueous ZIBs.

It is true that many gel electrolytes have been developed for ZIBs. The shortcomings and research gaps are the small overall capacity of the batteries, which may hinder their application in ZIBs as energy-storage devices. **In this work, we mainly focus on large-scale energy storage of the aqueous ZIBs.**

The reviewer's comments motivated us to add more background information on polymer gel electrolytes in ZIBs in general. We further compare the advantages and disadvantages of the CarraChi with previously reported hydrogel electrolytes in **Table R1** and add related references in the Revised Manuscript. It is seen that our work with the new battery design and the CarraChi gel electrolyte presents a much improved overall capacity for use as large-scale energy-storage batteries.

Table R1. Comparison of cells using gel electrolytes with our work.

Gel type	Tensile strength (MPa)	Ionic conductivity (mS cm ⁻¹)	t _{zn}	Current density (mA cm ⁻²)	Capacity (mAh cm ⁻²)	References
CarraChi	112	5.3	0.52	10	35/120	Our work
CMC/PNiPAM	37.9	0.17	0.54	5	1	1
P(ICZn-AAm)	0.12	2.15	0.93	0.25	-	2
PAM-PGO	-	31	-	3.5	1	3
PZHE	-	32	0.656	3	-	4
ILZE	9.12	16.9	-	0.2	0.5	5
SPE	-	19.6	0.7	0.1	0.5	6
Alg-Zn	-	18.3	0.75	1.77	0.885	7
PAMPSZn	-	20	0.4	1	1	8
TA-SA	-	24.2	0.74	1.13	-	9
TPU	3.5	19.8	-	10	10	10
Zn-SHn	-	34	-	10	5	11
PASHE	0.028	32.9	0.84	20	10	12

References

1. Dueramae I, Okhawilai M, Kasemsiri P, Uyama H, Kita R. Properties enhancement of carboxymethyl cellulose with thermo-responsive polymer as solid polymer electrolyte for zinc ion battery. *Sci. Rep.* 10, 12587 (2020).

2. Chan CY, Wang Z, Li Y, Yu H, Fei B, Xin JH. Single-Ion Conducting Double-Network Hydrogel Electrolytes for Long Cycling Zinc-Ion Batteries. *ACS Appl. Mater. Interfaces* 13, 30594-30602 (2021).
3. Abbasi A, et al. Phosphonated graphene oxide-modified polyacrylamide hydrogel electrolytes for solid-state zinc-ion batteries. *Electrochim. Acta* 435, 141365 (2022).
4. Leng K, et al. A Safe Polyzwitterionic Hydrogel Electrolyte for Long-Life Quasi-Solid State Zinc Metal Batteries. *Adv. Funct. Mater.* 30, 2001317 (2020).
5. Ma LT, et al. Hydrogen-Free and Dendrite-Free All-Solid-State Zn-Ion Batteries. *Adv. Mater.* 32, 1908121 (2020).
6. Ma LT, Chen SM, Li XL, Chen AO, Dong BB, Zhi CY. Liquid-Free All-Solid-State Zinc Batteries and Encapsulation-Free Flexible Batteries Enabled by In Situ Constructed Polymer Electrolyte. *Angew Chem., Int. Ed.* 59, 23836-23844 (2020).
7. Tang Y, et al. Ion-confinement effect enabled by gel electrolyte for highly reversible dendrite-free zinc metal anode. *Energy Storage Mater.* 27, 109-116 (2020).
8. Cong JL, et al. Ultra-stable and highly reversible aqueous zinc metal anodes with high preferred orientation deposition achieved by a polyanionic hydrogel electrolyte. *Energy Storage Mater.* 35, 586-594 (2021).
9. Zhang B, et al. Tuning Zn²⁺ coordination tunnel by hierarchical gel electrolyte for dendrite-free zinc anode. *Sci. Bull.* 67, 955-962 (2022).
10. Liu Q, Wang Y, Hong X, Zhou R, Hou Z, Zhang B. Elastomer-Alginate Interface for High-Power and High-Energy Zn Metal Anodes. *Adv. Energy Mater.* 12, 2200318 (2022).
11. Yang JL, Li J, Zhao JW, Liu K, Yang P, Fan HJ. Stable Zinc Anode Enabled by Zincophilic Polyanionic Hydrogel Layer. *Adv. Mater.* 34, 2202382 (2022).
12. Zhang W, et al. Kinetics-Boosted Effect Enabled by Zwitterionic Hydrogel Electrolyte for Highly Reversible Zinc Anode in Zinc-Ion Hybrid Micro-Supercapacitors. *Adv. Energy Mater.* 12, 2202219 (2022).

Updates in the Revised Manuscript:

We have added the comparison of cells using gel electrolytes with our work in Supplementary Table 1 (Supplementary Information). We have added the following discussions in the Revised Manuscript:

Line 45-47, Page 3: “Many efforts have been devoted to addressing these problems, such as using electrolyte additives, highly concentrated electrolytes,¹¹ nonaqueous electrolytes,¹² hydrogel electrolytes,¹³⁻¹⁵ 3D Zn anodes, and anode/electrolyte interface modification;¹⁶⁻²²”

Line 58-60, Page 4: “Hydrogel electrolytes show potential in addressing these issues; however, most of the related studies are still limited to small cell size and capacity, not practical for application (Supplementary Table 1).³³⁻³⁵”

References

12. Kao-ian W, Mohamad AA, Liu WR, Pornprasertsuk R, Siwamogsatham S, Kheawhom S. Stability Enhancement of Zinc-Ion Batteries Using Non-Aqueous Electrolytes. *Batteries Supercaps* **5**, 202100361 (2022).
13. Chan CY, Wang Z, Li Y, Yu H, Fei B, Xin JH. Single-Ion Conducting Double-Network Hydrogel Electrolytes for Long Cycling Zinc-Ion Batteries. *ACS Appl. Mater. Interfaces* **13**, 30594-30602 (2021).
14. Dueramae I, Okhawilai M, Kasemsiri P, Uyama H, Kita R. Properties enhancement of carboxymethyl cellulose with thermo-responsive polymer as solid polymer electrolyte for zinc ion battery. *Sci. Rep.* **10**, 12587 (2020).
15. Abbasi A, et al. Phosphonated graphene oxide-modified polyacrylamide hydrogel electrolytes for solid-state zinc-ion batteries. *Electrochimica Acta* **435**, 141365 (2022).
33. Liu Q, et al. Steric Molecular Combing Effect Enables Ultrafast Self-Healing Electrolyte in Quasi Solid-State Zinc-Ion Batteries. *ACS Energy Lett.* **7**, 2825–2832 (2022).
34. Liu Q, Wang Y, Hong X, Zhou R, Hou Z, Zhang B. Elastomer-Alginate Interface for High-Power and High-Energy Zn Metal Anodes. *Adv. Energy Mater.* **12**, 2200318 (2022).
35. Wang J, et al. Flexible and Anti-Freezing Zinc-ion Batteries Using a Guar-gum/Sodium-alginate/Ethylene-glycol Hydrogel Electrolyte. *Energy Storage Mater.* **41**, 599–605 (2021).

3. The authors refer Zn/CarraChi/Zn battery and Zn/GF/Zn battery. These are not batteries; they are symmetrical cells.

Reply: We thank the reviewer for pointing out this error. The Zn|CarraChi|Zn and Zn|GF|Zn are symmetric cells. We have corrected the description in the Revised Manuscript.

4. The concept of replenishing the electrolyte is fine. However, it needs to be clarified when and how much to fill the electrolyte. Also, the EIS of the system during cycling should be monitored.

Reply: We thank the reviewer for this helpful suggestion. Water or electrolyte (2M ZnSO₄ aqueous solution) is supplemented when the overpotential increases substantially in comparison to the initial overpotential. 0.5 ml g⁻¹ (ratio of water/2M ZnSO₄ to the active ZVO) of electrolyte was injected each time. The timing and amount of electrolyte filling, however, may vary depending on the cell size and environmental conditions, such as temperature and humidity. We have added the experimental detail in the Methods section.

We conducted EIS before and after cycling for 4000 h. It is seen that the R_{ct} decreases to some extent after cycling, which is ascribed to the improved reaction kinetics during cycling (Fig. R1). This further certifies the normal operation of the pouch symmetric cell.

Fig. R1. EIS curves of the Zn|CarraChi|Zn symmetric cell before and after cycling for 4000 h.

Updates in the Revised Manuscript:

We have added the EIS curves of Zn|CarraChi|Zn symmetric cells before and after cycling in the Revised Manuscript (**Fig. 4g**). We have also added the following discussion in the Revised Manuscript:

Line 217-219, Page 13: “The electrochemical impedance spectroscopy (EIS) curves before and after cycling also confirm that no short circuit occurred during cycling (Fig. 4h).”

Line 345-347, Page 20: “The water refilling procedure was conducted when the overpotential increased obviously in comparison to the initial overpotential, and 0.5 ml g⁻¹ (ratio of electrolyte to the active ZVO) electrolyte was injected every time. The battery is subjected to a pressure of 370 kPa during the testing process.”

5. It is unclear how tiny hydrogen bubbles leave the zinc surface, mainly when the viscous gel electrolyte covers the surface. Can some of these tiny hydrogen bubbles possibly accumulate on the surface and isolate the zinc surface from the electrolytes?

Reply: We thank the reviewer for this interesting question. To observe how the hydrogen bubbles form and accumulate on the interface between the Zn and gel electrolyte during the plating process, we carried out in situ optical microscope observation during Zn deposition (10 mA cm^{-2}). However, during the Zn deposition process, we did not find obvious bubbles accumulating on the Zn surface or isolating Zn from the gel electrolyte (Fig. R2).

If there were any accumulation of H_2 bubbles or Zn separation from the gel electrolyte, we should have found non-uniform Zn deposition or increased interface resistance after cell cycling. On the contrary, SEM images show uniform Zn morphology (Fig. 4h), and the R_{ct} after cycling is reduced (Fig. R1), indicating that there were no H_2 bubbles accumulating on the surface or isolating the Zn surface from the electrolyte. We speculate that any generated H_2 would diffuse and escape from the open cells before accumulation.

Fig. R2. In situ optical microscope images of Zn foil during plating at 10 mA cm^{-2} , without finding H_2 bubbles.

Updates in the Revised Manuscript:

We have added the in situ optical microscope images of Zn foil during plating in Supplementary Fig. 11 (Supplementary Information). We have added the following discussions in the Revised Manuscript:

Line 154-156, Page 9: “In situ optical microscope images further reveal the good suppression of the HER in the competing reactions between Zn deposition and HER on the Zn anode (Supplementary Fig. 11)”

6. It is essential to elaborate on the effects of the gel polymer electrolyte regarding the mitigation of zinc surface passivation by zinc hydroxide sulfate.

Reply: We thank the reviewer for the helpful suggestion. To study the effects of CarraChi regarding the mitigation of zinc surface passivation by zinc hydroxide sulfate, we conducted a soaking experiment by leaving the assembled symmetric cells with or without CarraChi for 1 week to investigate possible Zn corrosion. The experimental results show that the Zn in the Zn|CarraChi|Zn has a much weaker peak intensity of zinc hydroxide sulfate by-products in the XRD patterns (Fig. R3a). SEM images also reveal that a smooth surface of the Zn foil can be observed with CarraChi (Fig. R3b). In contrast, the Zn foil soaked in 2M ZnSO₄ solutions is covered with dendritic by-products of zinc hydroxide sulfate (Fig. R3c). Therefore, the CarraChi gel has a positive effect on mitigating the zinc hydroxide sulfate byproducts.

Fig. R3. a, XRD patterns of Zn foil in contact with ZnSO₄ liquid electrolyte and CarraChi gel electrolyte. SEM images of the Zn foil after standing for 1 week: b, with the CarraChi, and c, with ZnSO₄ liquid electrolyte.

Updates in the Revised Manuscript:

We have added the XRD patterns and SEM images of Zn foil soaked with CarraChi gel electrolyte and ZnSO₄ in the revised Supplementary Information (Supplementary Fig. 12). We have also added the following discussion in the Revised Manuscript to elaborate on the effect of CarraChi on the interface stability:

Line 156-160, Page 9: “The CarraChi gel electrolyte also reduces corrosion reactions between the Zn foil and ZnSO₄, which is proved by the weaker peak intensity of by-products in the X-ray diffraction (XRD) patterns and the smooth surface in the SEM images in the anti-corrosion experiment compared to the Zn anode with liquid electrolyte and GF separator (Supplementary Fig. 12).”

7. The capacity of the cell is measured to the cut-off voltage approaching 0 V, which is not practical for real applications. Anyway, the authors should compare their results with other results reported for liquid electrolytes of vanadium-oxide-based cells.

Reply: We thank the reviewer for the helpful suggestion. V₂O₅-based cathodes have been widely studied in the literature due to their high specific capacity. We find it is a common practice to apply the voltage range of 0.2–1.6 V for the V₂O₅-based cathode (Nat. Energy 2016, 1, 16119; ACS Appl. Mater. Interfaces, 2021,13, 30594-30602).

According to the reviewer’s comment, we compared the electrochemical performance of our refillable Zn batteries with previously reported vanadium-oxide-based cells in the literature (using liquid electrolytes) in Table R2. Our pouch cell with the CarraChi gel electrolyte has a maximum capacity of 0.9 Ah and excellent cycling stability, retaining 84% capacity after 200 cycles, which is much better than the previously reported pouch full cells.

Table R2. Comparison of the cyclic performance of pouch full cells with previous reports.

Cathode	Electrolyte	Capacity (Ah)	Current density	Cycle number	Retention (%)	Reference
ZVO	CarraChi	0.9	0.2 A g ⁻¹	200	84	Our work
VOH	Sulfolane–H ₂ O	0.2	0.5 A g ⁻¹	55	81.6	20
ZnVOH	ZnSO ₄	0.02	0.1 mA cm ⁻²	200	70.7	21
NH ₄ V ₄ O ₁₀	Zn(CF ₃ SO ₃) ₂	0.015	0.4 A g ⁻¹	240	83	22
KVP	Zn(CF ₃ SO ₃) ₂	0.0065	0.1 A g ⁻¹	60	98.5	23
HVO	ZOT-H ₂ O-PEG	0.0035	0.5 A g ⁻¹	300	81.7	24
V ₂ O ₅	PSIC gel	0.017	1.2 mA cm ⁻²	500	87.8	25

References

- Li M, et al. Comprehensive H₂O Molecules Regulation via Deep Eutectic Solvents for Ultra-Stable Zinc Metal Anode. *Angew Chem., Int. Ed.*, 62, 202215552 (2023).
- Lin Y, Mai Z, Liang H, Li Y, Yang G, Wang C. Dendrite-free Zn anode enabled by anionic surfactant-induced horizontal growth for highly-stable aqueous Zn-ion pouch cells. *Energy Environ. Sci.*, doi: 10.1039/d2ee03528f, (2023).
- Zhang H, et al. Inducing the Preferential Growth of Zn (002) Plane for Long Cycle Aqueous Zn-Ion Batteries. *Adv. Energy Mater.* 13, 2203254 (2022).
- Yang X, Deng W, Chen M, Wang Y, Sun CF. Mass-Produced, Quasi-Zero-Strain, Lattice-Water-Rich Inorganic Open-Frameworks for Ultrafast-Charging and Long-Cycling Zinc-Ion Batteries. *Adv. Mater.* 32, 2003592 (2020).
- Chen Y, et al. Low Current-Density Stable Zinc-Metal Batteries Via Aqueous/Organic Hybrid Electrolyte. *Batteries Supercaps* 5, 202200001 (2022).
- Chen Z, et al. Polymeric Single Ion Conductors with Enhanced Side Chains Motion for High-performance Solid Zinc Ion Batteries. *Adv. Mater.* 34, 2207682 (2022).
- Ma G, et al. Reshaping the electrolyte structure and interface chemistry for stable aqueous zinc batteries. *Energy Storage Mater.* 47, 203-210 (2022).

Updates in the Revised Manuscript:

We have added the performance comparison table (Table R2) in the revised Supplementary Information (**Supplementary Table 7**). We have also added the following discussion in the Revised Manuscript:

Line 270-274, Page 16: “The pouch full cell with the CarraChi gel electrolyte and open configuration has an initial capacity of 0.9 Ah and excellent cycling stability, retaining 84% capacity after 200 cycles, superior to full cells using liquid electrolytes and GF separator (Fig. 5f and Supplementary Fig. 22). This performance is also much better than that of the previously reported pouch full cells (Supplementary Table 7).”

8. Recently, several works have proposed different polymer gel electrolytes; the authors must compare the advantages and disadvantages of these works ([10.1038/s41598-020-69521-x](https://doi.org/10.1038/s41598-020-69521-x), [10.1016/j.electacta.2022.141365](https://doi.org/10.1016/j.electacta.2022.141365), [10.1021/acsami.1c05941](https://doi.org/10.1021/acsami.1c05941)).

Reply: We thank the reviewer for the reviewer's helpful suggestion. We have carefully read these recommended papers and compared the advantages and disadvantages of these polymer gels with our work. We have summarized the characteristic performances of gel electrolytes for Zn batteries (including the reviewer-suggested ones) and ours in Table R1 (page 2 of the Response Letter) for comparison.

It is found that gel electrolytes present high Zn^{2+} transference numbers and good ionic conductivity (Table R1), which are the advantages of gel electrolytes and exactly the reason why we chose a gel electrolyte to demonstrate our refillable battery configuration. The disadvantage of the previously reported gel electrolytes mainly lies in the relatively low current density and capacity, which may not meet the requirements of practical Zn batteries. In contrast, we focus on practical Zn batteries with a refillable and large-format configuration, taking advantage of the gel electrolyte and enabling operation under high current densities and large capacities.

Updates in the Revised Manuscript:

We have added the table of property comparison in the revised Supplementary Information (Supplementary Table 1). We have also added the following discussion in the Revised Manuscript:

Line 58-60, Page 4: “Hydrogel electrolytes show potential in addressing these issues; however, most of the related studies are still limited to small cell size and capacity, not practical for application (Supplementary Table 1).”

Line 202-203, Page 12: “Such a high capacity, current density, and DOD are hardly possible for closed-system Zn batteries with GF separators.⁴⁹”

Reviewer #2: This manuscript proposed a new gel electrolyte that can minimize the problems in Zn metal electrodes, such as hydrogen evolution reaction (HER) and nonuniform Zn deposition reaction. The authors claimed that the proposed mixed gel & liquid electrolyte (CarraChi gel-ZnSO₄) can bond water molecules and reduce the HER, and possesses high mechanical strength to prevent the Zn dendrite issue. Also, they argued that the high transference number of Zn ions and the facilitated desolvation process of Zn ions in CarraChi gel-ZnSO₄ enhanced the Zn redox reaction. Zn batteries using the CarraChi gel-ZnSO₄ showed surprising performances for both half-cells and full-cells. It was quite clever that the authors made a hole in the pouch cell to refill the water and to let the gaseous side-reaction products easily be removed away. I also personally believe that there is no reason for aqueous batteries to follow the conventional closed-type configurations of alkali metal-based batteries considering that the aqueous batteries can retain air or humid conditions. So, I do think proposing a new configuration, although there was a small change, can encourage researchers in the relevant field to think more broadly about cell design. However, I see there are lots of ambiguous speculations which require precise explanations, and more detailed comparisons with the reference Zn batteries are necessary. In light of all these circumstances, a rigorous major revision of this paper is recommended. The comments are shown below. Without precise explanations of every comment, this paper should be published in other journals.

Reply: We thank the reviewer for recognizing the novelty of our work and providing professional suggestions. We particularly agree with the reviewer that “there is no reason for aqueous batteries to follow the conventional closed-type configurations of alkali metal-based batteries” and “a new configuration, although there was a small change, can encourage researchers in the relevant field to think more broadly about cell design”. We have modified the Introduction and Conclusions based on the reviewer’s comments to better highlight the importance of the new cell design.

According to the reviewer’s comments, we have conducted more experiments and added more discussion and detailed comparisons to demonstrate the hypothesis clearly. All the concerns have been considered seriously and addressed in the Revised Manuscript. Please see below our point-by-point responses.

1. First of all, why would the formation of hydrogen bonds between the water molecules and CarraChi gel-ZnSO₄ reduce the HER? I know it is a conventional argument typically used in aqueous battery systems, but I don't see the scientific background of this argument. Also, the part saying that "the decrease of the non H-bond contents indicates a weaker water activity" should be clearly explained. Perhaps, the authors' perspectives on changes in electronic structure, and experimental results on changes in solvation structure might be meaningful.

Reply: We thank the reviewer for the insightful comments.

It is true that the relation between hydrogen bonding (H-bonding), water activity, and the HER has not been well understood. We wish to explain based on the literature and our experimental results.

According to Grotthuss mechanism of proton transport, proton transfers by means of concerted cleavage and formation of O-H bonds in an H-bonding network. This transfer displaces one of the protons from its original molecule and initiates a cascade of similar displacements throughout the H-bonding network (like Newton's cradle). A contiguous H-bonding network facilitates Grotthuss proton conduction during redox reactions (Nat. Energy 2019, 4, 123–130).

The rapid diffusion of protons through the H-bond network would facilitate their transport to the Zn anode surface for HER (ACS Energy Lett. 2023, 8, 40–47). Thus, the formation of H-bond between the hydrogel electrolyte and free water molecules destructs the H-bond network between water molecules, impedes the transport of protons, and thus reduces the water activity and HER.

In our work, we found that the CarraChi gel electrolyte forms H bonds with the water molecules and breaks the H-bond network of water molecules, as evidenced by Raman spectra (Supplementary Fig. 9). We also demonstrated that the HER in the CarraChi gel electrolyte is effectively suppressed compared with the Zn anodes in the aqueous electrolyte by LSV and in situ optical microscope (Supplementary Fig. 10 and 11). Thus, the formation of H bonds between the water molecules and CarraChi gel electrolyte suppresses the HER, in good agreement with the above mechanism discussed in the literature.

In addition, we agree with the reviewer's comments that the solvation structure has a direct impact on desolvation and HER mitigation. To reveal the changes in the solvation structure of Zn^{2+} , we further measured the Zn nuclear magnetic resonance (^{67}Zn NMR) for the ZnSO_4 solution and CarraChi gel electrolyte (Fig. R4). The ^{67}Zn chemical shift of the CarraChi gel electrolyte is slightly higher than that of the ZnSO_4 electrolyte, indicating that the CarraChi gel molecules replace bound H_2O molecules in the optimized Zn^{2+} solvation sheath, which is in agreement with the Raman results. The changed solvation structure is considered a factor in decreasing the water activity and mitigating the HER.

Fig. R4. ^{67}Zn NMR spectra of ZnSO_4 aqueous solution and the CarraChi gel electrolyte.

Updates in the Revised Manuscript:

We have added the following discussion in the Revised Manuscript:

Line 149-156, Page 9: “In comparison to the ZnSO_4 aqueous solution, the proportion of strong H-bonding water reduces while weak H-bonding water increases in the CarraChi- ZnSO_4 , suggesting the bonding between the CarraChi gel and H_2O molecules and reconstruction of the H-bond network of free H_2O , which are expected to limit HER.^{29, 46, 47} The suppressed HER on the Zn anode with CarraChi is also proved by a much lower onset overpotential (-1.40 V vs. standard hydrogen electrode, SHE) than that with a GF separator (-1.26 V vs. SHE) (Supplementary Fig. 10). In situ optical microscope images further reveal the good suppression of the HER in the competing reactions between Zn deposition and HER on the Zn anode (Supplementary Fig. 11).”

2. The authors showed LSV data to claim that CarraChi gel-ZnSO₄ can suppress the HER. However, it is questionable why the authors obtained the overpotential at 5 mA cm⁻². Normally, we are interested in the on-set potential of HER. If the authors want to claim the kinetically suppressing effect of CarraChi gel-ZnSO₄ on HER in the way they used in the paper, it is necessary to show that the overpotential of the Zn redox reaction is lower than the reference case. Otherwise, the method the authors used will show the effects of low ionic conductivity (5.3 mS cm⁻¹, which is lower than the typically reported ionic conductivity of aqueous 2M ZnSO₄ electrolyte), which will obviously show a higher overpotential for both the Zn redox reaction and HER. By the way, you should change Supplementary Fig. 12 to Supplementary Fig. 10 on page 10, and Supplementary Fig. 10 to 11 on page 12.

Reply: We thank the reviewer for the constructive suggestion. We fully agree with the reviewer's comment that the onset potential of HER is important to evaluate the HER kinetics and the ionic conductivity will affect the overpotential of HER. Thus, instead of comparing the overpotential at 5 mA cm⁻², we compared the onset potential of HER in symmetric cells. To eliminate the effect of ohm resistance on the measurement results, we carried out iR correction for the LSV curves and calculated the onset potential for HER (Fig. R5), which was according to the method in previous reports (Nat. Commun. 2019, 10, 1348; Sci. Adv. 2015, 1, e1500259) (the onset potential is the point where the oblique line extension intersects the line parallel to the horizontal axis). In particular, the battery with the CarraChi exhibits a more negative onset potential of -1.40 V, in comparison to that with the GF separator (-1.26 V), indicating the effective mitigation of HER.

Also, we very much thank you for your kind corrections to the figure indexes and have checked these indexes from Fig. 10 to Fig. 11 in the Supplementary Information and corrected them.

Fig. R5. LSV curves of Zn with/without CarraChi with iR correction.

Updates in the Revised Manuscript:

We have updated the LSV curves and onset potentials in the revised Supplementary Information (Supplementary Fig. 10). We also corrected the figure index errors from Fig. 10 to Fig. 11. We have also added the following discussion in the Revised Manuscript:

Line 152-154, Page 9: “The suppressed HER on the Zn anode with CarraChi is also proved by a much lower onset overpotential (−1.40 V vs. standard hydrogen electrode, SHE) than that with a GF separator (−1.26 V vs. SHE) (Supplementary Fig. 10).”

3. Where did the C-N-C bond come from in Supplementary Fig. 3b? Also, I believe Fig. 2g needs to be revised. In the manuscript, it is said that Fig. 2g is about the FT-IR result comparing the spectrum of k-carrageenan and chitosan with CarraChi. However, there are only spectra of CarraChi and CarraChi gel-ZnSO₄ in Fig. 2g. As a result, I could not clearly observe the red shift of the -OH/-NH₂ peak in CarraChi.

Reply: We thank the reviewer for the helpful comments. Following the reviewer’s comments, we double-checked the peak information of N species in Supplementary Fig. 3 and corrected the bond information. The peak located at 399.5 eV is ascribed to the C-N bond (Fig. R6b).

We revised **Fig. 2g** and added the FTIR curves of CarraChi gel, chitosan, and k-carrageenan in the updated figure. In comparison to the chitosan and k-carrageenan, a clear red shift of the -OH/-NH₂ peak can be observed from 3256.7 to 3136.3 cm⁻¹ in CarraChi (Fig. R7).

Fig. R6. **a**, XPS full spectrum, and **b**, N1s spectra, and **c**, S2p spectra of the CarraChi gel.

Fig. R7. FTIR spectra of the CarraChi gel, chitosan, and k-carrageenan.

Updates in the Revised Manuscript:

We have corrected the bond information of the N1s in Supplementary Fig. 3 in the revised Supplementary Information. We have also revised the FTIR curves in Fig. 2g and added related discussions in the Revised Manuscript:

Line 102-104, Page 6, “In the FTIR spectrum of CarraChi gel, the peaks at 1558 and 1449 cm⁻¹ are respectively ascribed to the N-H and C-N stretching vibrations in chitosan,³⁷”

4. I am not sure if the part saying the “high mechanical strength of CarraChi gel-ZnSO₄ can suppress the dendritic growth of Zn metal” is really reasonable. It is known that the shear modulus of the surface

layer should be at least 2 times higher than that of the metal electrode to mechanically induce uniform metal deposition reaction and prevent penetration.

Reply: We thank the reviewer for the helpful comments. According to the widely cited Monroe-Newman model, which was theoretically simulated in 2005 (J. Electrochem. Soc. 2005, 152, 396–404), Li dendrite growth can be successfully suppressed if the shear modulus of solid-state electrolytes is 2-fold larger than that of Li (4.8 GPa at 298 K). However, this conclusion is based on the theoretical calculation with an ideal model. Archer et al. reported that polymer electrolytes with a shear modulus of the same order of magnitude as Li could also suppress Li dendrite growth (J. Am. Chem. Soc. 2014, 136, 7395-7402). In general, mechanical strength is not the only factor that affects dendrites but improving the mechanical strength should be beneficial to suppress metal dendrites. Therefore, we have revised the explanation in the Revised Manuscript.

Updates in the Revised Manuscript:

Line 112-114, Page 6: “Such an improved mechanical property of the CarraChi gel electrolyte is expected to mitigate Zn dendrite penetration in the charge/discharge process.”⁴²”

5. In Supplementary Fig. 6, I think it was too short to observe true steady state current for both Zn|CarraChi|Zn and Zn|GF|Zn. Also, I think it is really confusing that the authors use both the terms CarraChi and CarraChi gel-ZnSO₄ throughout the paper. As far as I understood, it should be CarraChi gel-ZnSO₄. Am I right?

Reply: We thank the reviewer for the helpful suggestion. We remeasured the chronoamperometry of the Zn|CarraChi|Zn and Zn|GF|Zn and extended the test time from 15 to 35 minutes according to literature (Nat. Commun. 2022, 13, 5348). We further recalculated the t_{Zn} , and the value of t_{Zn} is 0.52 for Zn|CarraChi|Zn and 0.29 for Zn|GF|Zn symmetric cells (Fig. R8).

The reviewer is right that it should be the CarraChi-ZnSO₄ gel electrolyte in the batteries. However, this abbreviation is too long in figures. Thus, we use the term *CarraChi* or *CarraChi-ZnSO₄* for the CarraChi-ZnSO₄ gel electrolyte and *CarraChi membrane* or *CarraChi gel* for a pure gel without

ZnSO₄. All the terms have been defined clearly and used consistently in the Revised Manuscript to avoid confusion.

Fig. R8. Chronoamperometry curves with a constant voltage polarization of 20 mV. The insets show the EIS curves before and after polarization. **a**, Zn|CarraChi|Zn. **b**, Zn|GF|Zn.

Updates in the Revised Manuscript:

We have further updated the chronoamperometry and EIS curves in Supplementary Fig. 6 in the revised Supplementary Information. We have also updated the term *CarraChi* or *CarraChi-ZnSO₄* for the CarraChi-ZnSO₄ gel electrolyte and *CarraChi membrane* or *CarraChi gel* for a pure gel without ZnSO₄ and other discussions in the Revised Manuscript.

Line 129-133, Page 8: “The Zn²⁺ transference numbers (t_{Zn}) of CarraChi and GF separator have also been obtained using the Bruce-Vincent method,⁴³ and the CarraChi gel electrolyte has a high t_{Zn} of 0.52, which exceeds that of the aqueous electrolyte in commercial GF (0.29), indicating the superior Zn²⁺ transfer property in CarraChi gel electrolyte (Supplementary Fig. 6, Supplementary Table 3).”

6. Please provide the zeta potential of GF in a neutral solution before and after the addition of ZnSO₄.

Reply: We thank the reviewer for the helpful suggestion. Following the comment, we tested the zeta potential of pulverized GF in a neutral solution before and after the addition of ZnSO₄. The zeta potential of GF powder is -26.4 mV (Fig. R9a). After adding ZnSO₄, a negligible shift of the zeta

potential is observed in GF, indicating that there is no interaction between the Zn^{2+} and GF powders, very different from the interaction between the Zn^{2+} and CarraChi (Fig. R9b).

Fig. R9. Zeta potential: **a**, the broken GF powder in a neutral solution before and after the addition of $ZnSO_4$; **b**, CarraChi gel before and after the addition of $ZnSO_4$.

Updates in the Revised Manuscript:

We have added the zeta potential curves of GF powder in Supplementary Fig. 7 (Supplementary Information). We have also added a related discussion in the Revised Manuscript.

Line 136-139, Page 8: “After the addition of $ZnSO_4$, the less negative zeta potential of -9.4 mV reveals the effective adsorption or crosslinking of Zn^{2+} by the CarraChi, facilitating the desolvation of Zn^{2+} during the deposition process, which differs from GF separator without adsorption toward Zn^{2+} (Supplementary Fig. 7).

7. The authors say a high transference number of Zn ions, and a fast desolvation process for Zn ions in CarraChi gel- $ZnSO_4$ can result in faster reaction kinetics of Zn metal electrodes compared to the GF case. It seems true since the R_{ct} of Zn/Zn cell using CarraChi gel- $ZnSO_4$ is lower than that of the cell using GF. However, the overpotential of the Zn/Zn cell using CarraChi gel- $ZnSO_4$ is much higher than that of the cell using GF. Is this due to the low ionic conductivity of CarraChi gel- $ZnSO_4$? Also, the authors claimed that using CarraChi gel- $ZnSO_4$ can increase the CE of Zn/Cu cells. Due to some error points that show abnormal CEs for CarraChi gel- $ZnSO_4$ cell, the average CE of cell using CarraChi

gel-ZnSO₄ might be similar or a little higher than the cell of GF, but it seems like Zn/Cu cell using GF has higher CEs in most of the cycles before the 150th cycle. Please provide reasonable explanations for these circumstances.

Reply: We thank the reviewer for the helpful comments. We wish to clarify below:

Regarding “Is this due to the low ionic conductivity of CarraChi gel-ZnSO₄?”

We agree with the reviewer’s point that the Zn/Zn cell using CarraChi shows a higher overpotential than that using a GF separator mainly because the ionic conductivity of CarraChi is low than the liquid electrolyte. To understand the cause of lower R_{ct} and higher overpotential in symmetric cells, we conducted an extensive review of the literature about hydrogel electrolytes for ZIBs. Most previous papers (for example, Adv. Mater. 2022, 34, 2202382; Energy Storage Mater. 2022, 46, 523–534) ascribe lower R_{ct} to suppressing corrosion reaction and higher overpotential to lower ionic conductivity. We carried out more experiments to verify this explanation.

To reveal the effect of corrosion reaction on R_{ct} , we carried out the EIS measurement for Zn|GF|Zn symmetric cells with different conditions, including fresh cells, after standing for 7 days, and after cycling (Fig. R10). A much higher R_{ct} is obtained after standing for 7 days, demonstrating the occurrence of severe side reactions. The XRD patterns and SEM images further show that more by-products formed in Zn|GF|Zn (Fig. R11). These results certify that the R_{ct} is profoundly affected by the corrosion reactions.

Moreover, in comparison to the R_{ct} after standing for 7 days, a much smaller R_{ct} for Zn|GF|Zn symmetric cells is obtained after cycling, indicating the suppression of corrosion reactions during electrochemical cycling. Therefore, we believe that the overpotential is mainly determined by the ionic conductivity in the symmetric cell.

Fig. R10. EIS curves of the Zn symmetric cell with GF separator with different conditions including without standing, standing for 7 days, and after cycling (1 mA cm^{-2} and 1 mAh cm^{-2} for 20 cycles)

Fig. R11. **a**, XRD patterns of Zn foil in contact with ZnSO_4 liquid electrolyte and CarraChi gel electrolyte. SEM images of the Zn foil after standing for 1 week: **b**, with the CarraChi, and **c**, with ZnSO_4 liquid electrolyte and the GF separator.

Regarding “Zn/Cu cell using GF has higher CEs in most of the cycles before the 150th cycle”

Although it seems like Zn/Cu cell using GF has higher CEs in most of the cycles before the 150th cycle, a much higher initial CE is achieved for CarraChi (92.7%) in comparison to GF (83.8%). Therefore, the Zn|CarraChi|Cu exhibits a comparable average CE with Zn|GF|Cu before the 150th cycle. We have conducted more tests for the CE of a Zn/Cu cell using CarraChi, which is as high as 99.8% after 300 cycles (Fig. R12). The curves have also been updated in the Revised Manuscript.

Fig. R12. CE curves of Zn plating/stripping in Zn|CarraChi|Cu and Zn|GF|Cu cells at 5 mA cm^{-2} with a capacity of 1 mAh cm^{-2} .

Updates in the Revised Manuscript:

We have added the XRD patterns and SEM images of Zn foil soaked with CarraChi gel electrolyte and ZnSO_4 in the revised Supplementary Information (**Supplementary Fig. 12**). We have updated the CE curves and data in **Fig. 4d** in the Revised Manuscript.

Line 207-209, Page 12: “Moreover, in comparison to the unstable Zn plating/stripping less than 200 cycles with GF separator, the Zn anode with CarraChi gel electrolyte displays a stable CE for 600 cycles at 5 mA cm^{-2} , which is as high as 99.8% after 300 cycles (Fig. 4d and 4e).”

Line 156-160, Page 9: “The CarraChi gel electrolyte also reduces corrosion reactions between the Zn foil and ZnSO_4 , which is proved by the weaker peak intensity of by-products in the X-ray diffraction (XRD) patterns and the smooth surface in the SEM images in the anti-corrosion experiment compared to the Zn anode with liquid electrolyte and GF separator (Supplementary Fig. 12).”

8. Please provide the current density condition used for comparing the SEM image in Fig. 3f, and Supplementary Fig. 9a, b).

Reply: We thank the reviewer for the helpful suggestion. The applied current density in Fig. 3f and Supplementary Fig. 14a, b is 10 mA cm^{-2} .

Updates in the Revised Manuscript:

We have updated the applied current density in the Revised Manuscript and Supplementary Information.

Line 190-191, Page 11: “f, top, and g, side views with the CarraChi gel electrolyte (10 mA cm⁻² with deposition capacity of 10 mAh cm⁻²);”

Page S15, “Supplementary Fig. 14 | a-d, SEM images of bare Zn deposited at 10 mA cm⁻² with different capacities. a, 20 mAh cm⁻² and b, 30 mAh cm⁻² in Zn|CarraChi|Zn; c, 20 mAh cm⁻² and d, 30 mAh cm⁻² in Zn|GF|Zn.”

9. Why is the rate capability of the full cell using CarraChi gel-ZnSO₄ better than the reference case? I believe using CarraChi gel-ZnSO₄ can impinge the overpotential as is proved in Zn/Zn cell test. Please also provide the voltage profile of the Zn|GF|ZVO and compare it in Fig. 5a.

Reply: We thank the reviewer for the helpful comments. We wish to clarify below.

Regarding “the better rate performance of Zn|CarraChi|ZVO”

It is true that a higher overpotential is observed for Zn|CarraChi|Zn. However, the transport kinetics of Zn²⁺ should also be considered when assembling with the ZVO cathode. The higher t_{Zn} of the CarraChi gel electrolyte ensures an effective Zn²⁺ supplement during the high-rate charge/discharge process. To reveal the transport kinetics of Zn²⁺, we carried out the EIS measurements for the full cells before and after cycling (Fig. R13). A smaller R_{ct} is obtained for Zn|CarraChi|ZVO, demonstrating the improved redox kinetics and stable interface charge transfer even under high currents.

Fig. R13. EIS curves of Zn|CarraChi|ZVO and Zn|GF|ZVO: **a**, before cycling and **b**, after cycling.

In addition to the decreased kinetics, dissolution of the ZVO cathode in 2M ZnSO₄ (pH=4.3) may also contribute to the lower rate of Zn|GF|ZVO during the cycling, which can be evidenced by the yellow dissolved ZVO on the GF separator (Fig. R14).

Fig. R14. Optical images of CarraChi and GF after standing with ZVO cathode for 1h (the dissolved ZVO absorbed on GF while there is no dissolved ZVO on CarraChi).

Regarding “comparing the voltage profile of the Zn|GF|ZVO”

The voltage profile of the Zn|GF|ZVO is now provided in Fig. R15. The Zn|CarraChi|ZVO cell shows a higher overpotential than that of the Zn|GF|ZVO cell at 0.2 A g⁻¹, but much lower overpotential than

the Zn|GF|ZVO cell from 0.5 to 4 A g⁻¹, demonstrating the improved redox kinetics of Zn|CarraChi|ZVO under higher current density (Fig. R15).

Fig. R15. Voltage profiles of (a) Zn|GF|ZVO and (b) Zn|CarraChi|ZVO at various charge/discharge current densities.

Updates in the Revised Manuscript:

We have provided the voltage profiles of Zn|GF|ZVO in Supplementary Fig. 18 in the Revised Supplementary Information. We have also provided the optical images of CarraChi and GF after standing with ZVO cathode in Supplementary Fig. 19.

Line 249-250, Page 15: “The superior cycling stability due to effective suppression of dendrite and ZVO dissolution ensures the long life of the Zn batteries (Supplementary Fig. 19).”

10. Lastly, please compare the refillable pouch cell test result in Fig. 4g and Fig. 5f with the GF cells.

Reply: We thank the reviewer for the helpful suggestion. We first tested the refillable pouch cell with a GF separator. For the Zn|GF|Zn pouch symmetric cell, the short circuit occurs when the capacity increases to 2 mAh cm⁻² at 10 mA cm⁻² in the pouch cell (Fig. R16a), which is due to the dendrite piercing into the porous GF, proved by the EIS curves (Fig. R16b). This is in sharp contrast to the stable cycling of Zn|CarraChi|Zn pouch symmetric cell with 35 mAh cm⁻².

Fig. R16. a, Cycling performance of the symmetric Zn|GF|Zn pouch cell with various capacities from 1 to 10 mAh cm⁻² at 10 mA cm⁻². **b**, EIS curve of the Zn|GF|Zn pouch cell after cycling.

We also tested the cycling stability of Zn|GF|ZVO battery in a pouch cell as a control. A short circuit occurred after cycling for 23 cycles, which is also verified by the abnormal voltage curves and EIS result (Fig. R17 and Fig. R18). These results confirm the infeasibility of using the GF separator for large-format cells.

Fig. R17. Cycling performance of the pouch cells with CarraChi and GF.

Fig. R18. a, Voltage profiles and **b**, EIS curve of Zn|GF|ZVO pouch cell after the battery failure.

Updates in the Revised Manuscript:

We have provided the cycling performance of the symmetric Zn|GF|Zn pouch cell in Supplementary Fig. 16. We have also added the cycling performance of the Zn|GF|ZVO pouch cell in **Fig. 5f** in the Revised Manuscript. The voltage profile and EIS curves of the Zn|GF|ZVO pouch cell was also added in Supplementary Information (Supplementary Fig. 22). Related discussion has been added in the Revised Manuscript.

Line 222-225, Page 13: “In contrast, the Zn|GF|Zn pouch cell exhibits limited capacity, and a short circuit occurs when the specific capacity increases to 2 mAh cm⁻² (Supplementary Fig. 16), implying the unavailability of GF in the large-format refillable batteries.”

Line 270-274, Page 16: “The pouch full cell with the CarraChi gel electrolyte and open configuration has an initial capacity of 0.9 Ah and excellent cycling stability, retaining 84% capacity after 200 cycles, superior to full cells using liquid electrolytes and GF separator (Fig. 5f and Supplementary Fig. 22). This performance is also much better than that of the previously reported pouch full cells (Supplementary Table 7).”

Reviewer #3: The present manuscript proposed a hydrogel that can be prepared on a large scale and a refillable battery design. Indeed, the overall electrochemical performance of the ZIBs has been improved in this study. However, the work fails to reflect the big progress in the development of high-performance ZIBs. Many studies proposed hydrogel systems. The current work just selected a different raw material that has been commercialized. The proposed battery design also disables addressing the key challenge of ZIBs, such as HER and long-cycling stability of the cathode. In all, the proposed concept of the scalable synthesis of hydrogel and battery design cannot present an impressive innovation. Thus, I do not support the acceptance of the manuscript for publication in the present form.

Reply: We thank the reviewer for the positive comment that “the overall electrochemical performance of the ZIBs has been improved in this study”. We respectfully disagree that “the work fails to reflect the big progress in the development of high-performance ZIBs” and “just selected a different raw material”. **This work mainly presents a new battery design, namely the refillable and large-format configuration, for large-scale aqueous batteries with a great leap of performance forward, whereas the employed hydrogel material or its synthesis is not the core point.** We have to say sorry for not having explained the innovation point of this work very well in our previous version. We wish to clarify herein:

Regarding “the work fails to reflect the big progress in the development of high-performance ZIBs.”

The core point of the submitted work is the refillable and large-format configuration for the aqueous batteries which is enabled by the designed CarraChi gel electrolyte. This configuration aims to address issues including gas build-up, battery swelling, and electrolyte consumption in practical large-scale batteries.

With this cell design, Zn metal anodes have a greatly improved performance, realizing an areal capacity of 35 mAh cm^{-2} (DOD of 65%) and a record-high cumulative cycling capacity of 1286 Ah at 10 mA cm^{-2} in pouch cells, 100 times higher than the capacities of state-of-the-art Zn cells in the literature (generally $< 10 \text{ Ah}$, Fig. 4h). For Zn-ion full batteries, we have compared the electrochemical performance of the Zn|CarraChi|ZVO with previously reported pouch cells in Table R3 (Supplementary Table 7 in Supplementary Information). **Our ZIB shows a much-improved capacity**

and cycling life. We believe these results reflect the great progress in high-performance aqueous Zn batteries.

Table R3. Comparison of the cyclic performance of pouch full cells with previous reports.

Cathode	Electrolyte	Capacity (Ah)	Current density	Cycle number	Retention (%)	References
ZVO	CarraChi	0.9	0.2 A g ⁻¹	200	84	Our work
VOH	Sulfolane-H ₂ O	0.2	0.5 A g ⁻¹	55	81.6	20
ZnVOH	ZnSO ₄	0.02	0.1 mA cm ⁻²	200	70.7	21
NH ₄ V ₄ O ₁₀	Zn(CF ₃ SO ₃) ₂	0.015	0.4 A g ⁻¹	240	83	22
KVP	Zn(CF ₃ SO ₃) ₂	0.0065	0.1 A g ⁻¹	60	98.5	23
HVO	ZOT-H ₂ O-PEG	0.0035	0.5 A g ⁻¹	300	81.7	24
V ₂ O ₅	Gel	0.017	1.2 mA cm ⁻²	500	87.8	25
VOH	DME-Zn(OTF) ₂	0.062	-	100	83.5	26

References

- Li M, et al. Comprehensive H₂O Molecules Regulation via Deep Eutectic Solvents for Ultra-Stable Zinc Metal Anode. *Angew Chem., Int. Ed.*, 62, 202215552 (2023).
- Lin Y, Mai Z, Liang H, Li Y, Yang G, Wang C. Dendrite-free Zn anode enabled by anionic surfactant-induced horizontal growth for highly-stable aqueous Zn-ion pouch cells. *Energy Environ. Sci.*, doi: 10.1039/d2ee03528f, (2023).
- Zhang H, et al. Inducing the Preferential Growth of Zn (002) Plane for Long Cycle Aqueous Zn-Ion Batteries. *Adv. Energy Mater.* 13, 2203254 (2022).
- Yang X, Deng W, Chen M, Wang Y, Sun CF. Mass-Productible, Quasi-Zero-Strain, Lattice-Water-Rich Inorganic Open-Frameworks for Ultrafast-Charging and Long-Cycling Zinc-Ion Batteries. *Adv. Mater.* 32, 2003592 (2020).
- Chen Y, et al. Low Current-Density Stable Zinc-Metal Batteries Via Aqueous/Organic Hybrid Electrolyte. *Batteries Supercaps* 5, 202200001 (2022).
- Chen Z, et al. Polymeric Single Ion Conductors with Enhanced Side Chains Motion for High-performance Solid Zinc Ion Batteries. *Adv. Mater.* 34, 2207682 (2022).
- Ma G, et al. Reshaping the electrolyte structure and interface chemistry for stable aqueous zinc batteries. *Energy Storage Mater.* 47, 203-210 (2022).

Regarding “The current work just selected a different raw material that has been commercialized.”

We note that **the core innovation of this work is the refillable and large-format configuration for large-scale aqueous batteries. The use of the CarraChi gel is a means to realize our design** of the refillable and large-format battery configuration and address issues therein, including electrolyte leakage, fast water evaporation, lamination, and assembly of large pouch cells.

Therefore, this work does not intend to report a gel electrolyte. It is because of the new battery configuration enabled by the CarraChi gel electrolyte that we achieve the aforementioned practically-applicable high areal capacity (35 mAh cm^{-2}), current density, DOD (65%), cycle life ($>4000 \text{ h}$), record-high cumulative capacity ($1286 \text{ Ah at } 10 \text{ mA cm}^{-2}$), which are challenging to for Zn batteries using conventional liquid/gel electrolytes without this battery design.

Regarding “The proposed battery design also disables addressing the key challenge of ZIBs, such as HER and long-cycling stability of the cathode.”

The core innovation of our work is the refillable and large-format configuration for the aqueous batteries. Such a unique configuration addresses the key challenges in large-scale batteries including gas build-up, battery swelling, and electrolyte consumption. Meanwhile, the use of the CarraChi gel electrolyte is beneficial to suppress the dendrite growth, HER, and water evaporation, ensuring a long-term lifespan.

Regarding HER, HER is not likely to be eliminated completely because HER is thermodynamically favorable on the Zn metal surface in an aqueous environment. As we discuss further below, our CarraChi gel electrolyte reduces HER; our open-system battery also enables releasing any generated gas in large-format cells. As a result, the long-cycling stability of the full cells is much improved in large-format pouch cells. We have compared the performance of Zn|CarraChi|ZVO pouch cells with previously reported ZIB pouch cells in Table R3. It is seen that the Zn|CarraChi|ZVO pouch cell with our proposed battery design exhibits superior cycling stability under high capacity.

This new design realized significant advances in the performance of ZIBs, especially overall capacity. Such a refillable battery configuration may introduce a paradigm shift in understanding and designing aqueous batteries for large-scale energy storage.

In addition, we have carefully addressed all the reviewer's concerns in our point-by-point responses below. We have also revised our manuscript to better illustrate our idea and findings according to the reviewer's comments.

1. The authors proposed a hydrogel enabling to suppress the electrolyte loss. However, why do the authors need to refill the water? Refilling water means that the electrolyte will be consumed or can not be preserved very well in this system. In this situation, there is an obvious contradiction between the proposed concept and experimental results. Meanwhile, the authors should clarify why the electrolyte is consumed or can not be preserved even though hydrogel has been used in this manuscript. Does the refilled water affect the electrolyte concentration and further the electrochemical performance? If it does, it represents that the as-proposed battery design may fail to maintain the electrochemical performance. If it doesn't, the authors need to elaborately explain why the electrochemical performance does not change even though the electrolyte concentration changes after refilling the water. Additionally, as claimed that the hydrogel can suppress HER, but why do the authors still need to design a gas outlet?

Reply: We thank the reviewer for the helpful comments. Here are the answers to each of these questions.

Regarding “why do the authors need to refill the water?” and “contradiction between the proposed concept and experimental results”

This work intends to build large-format aqueous ZIBs, which are challenged by gas build-up, battery swelling, and electrolyte consumption over long-term cycling. We propose the open system to effectively solve the issues of gas build-up and battery swelling. **As water consumption is almost inevitable in large ZIBs over long-time cycling due to the thermodynamically favored HER, we solve the water consumption issue by refilling water.**

Due to the capability of the CarraChi gel with abundant polar functional groups (-OH, -NH₂, -SO₄²⁻) to bond H₂O molecules, **we use the CarraChi gel to preserve water, prevent electrolyte leakage and reduce water consumption.** However, water consumption is almost inevitable in large-format

ZIB cells over long-term cycling due to HER side reaction (even if it is greatly suppressed). **We take advantage of the open-system cell configuration to refill water into the cells, which is simple and cost-effective, so as to supplement any consumed water to ensure good performance and long life** (more than 4000 h in Fig. 4g).

Thus, using the proposed battery design and refilling water are not contradictory, but together contribute to high-performance, long-life, large-scale ZIBs.

Updates in the Revised Manuscript:

Line 65-67, Page 4: “The crosslinked k-carrageenan and chitosan (CarraChi) gel electrolyte has numerous polar functional groups (-OH, -NH₂, -SO₄²⁻) that bond water molecules to suppress fast electrolyte evaporation and the HER.”

Regarding “Does the refilled water affect the electrolyte concentration and further the electrochemical performance?”

We aimed to design the refillable and large-format configuration for the large-format Zn batteries. **Suitable water refilling can be an effective method to sustain the electrolyte concentration (ionic conductivity) of the CarraChi gel electrolyte in large-format cells.** In this work, the water refilling ensures good ionic conductivity and electrochemical cycling performance, which is proved by the long cycle life and slightly decreased overpotential after refilling the water or electrolyte in Fig. 4g.

It should be noted that evaporation of the water can lead to the salting-out of some ZnSO₄ in the CarraChi electrolyte, which is harmful to the Zn²⁺ transport. Refilling water helps redissolve the ZnSO₄ in the CarraChi, and maintains the proper concentration in the gel electrolyte. Moreover, considering the low cost of ZnSO₄ (~8.2 USD kg⁻¹), the ZnSO₄ solution can also be used to supplement the electrolyte consumption once the polarization potential increases markedly (Fig. 4g).

Updates in the Revised Manuscript:

Line 214-219, Page 12: “The pouch cell has an ultralong life of ~4000 h (65% DOD) with an areal capacity of 35 mAh cm⁻² at 10 mA cm⁻² (Fig. 4g). When the overpotential increased markedly due to electrolyte consumption during cycling, pure water or 2 M ZnSO₄ was refilled to sustain the ionic conductivity for normal operation, which is indicated in the voltage-time curves (Fig. 4g). The electrochemical impedance spectroscopy (EIS) curves before and after cycling also confirm that no short circuit occurred during cycling (Fig. 4h).”

Regarding “as claimed that the hydrogel can suppress HER, but why do the authors still need to design a gas outlet?”

In general, the HER cannot be completely eliminated because metallic Zn is thermodynamically unstable in aqueous environments. The HER inevitably occurs on the Zn-electrolyte interface in 2M ZnSO₄ electrolyte (pH=4.3). In our work, the designed CarraChi gel can mitigate the HER by destructing the H-bond network between water molecules, which are responsible for fast proton transport. Therefore, the HER can be suppressed by the CarraChi gel electrolyte significantly. However, we do not seek to completely eliminate HER in large-format ZIBs over long-term cycling (which we believe is unlikely). **Instead, we design the open-system battery with the gas outlet to avoid H₂ accumulation and battery swelling, thereby circumventing problems caused by HER, especially for large-scale, long-life batteries.**

2. K-carrageenan and chitosan have been successfully commercialized for many years. Therefore, large-scale preparation of hydrogel using these two raw materials is not an innovation of this manuscript.

Reply: We agree with the reviewer that the CarraChi gel can be prepared on a large scale. **This is an advantage but not the main innovation of this work. The core point of this work is the refillable, open-system battery configuration for large-scale aqueous batteries.**

The use of the CarraChi gel in our work is a means to realize the open, refillable, large-format battery configuration. We used the CarraChi gel because it can prevent electrolyte leakage, reduce water evaporation, and also homogenize the electric field distribution for dendrite-free Zn deposition.

Moreover, the low cost and large-scale preparation of CarraChi ensure the production of practical large-format Zn batteries for large-scale energy storage.

As a result, **our refillable cells with the CarraChi electrolyte achieve high capacity, current density, DOD, and ultralong lifespan**, which are not possible for ZIB with GF separators or any other previously reported hydrogel electrolytes with traditional sealed cell configurations. It is our new battery design that enables large-format, practically applicable ZIBs.

3. Figure 2h demonstrated that the tensile stress of Carrachi-ZnSO₄ has evidently decreased compared with CarraChi, suggesting that some chemical bonds have changed in this gel because of Zn²⁺. Please make a detailed explanation of what happened to the gel when Zn²⁺ was added. Authors should also supplement the visualization proof to demonstrate the gel can prevent dendrite penetration.

Reply: We thank the reviewer for the suggested comments.

Regarding “Please make a detailed explanation of what happened to the gel when Zn²⁺ was added.”

The bonding effect occurs between the CarraChi gel and Zn²⁺ after introducing Zn²⁺. It should be pointed out that the reduced tensile strength of the CarraChi-ZnSO₄ than the dry CarraChi is due to the existence of H₂O, as is evidenced by the tensile stress-strain curves of the dry and wet CarraChi gel (Fig. R19). To explain the bonding effect of CarraChi gel and Zn²⁺, we would better compare the CarraChi-ZnSO₄ to CarraChi-H₂O instead of dry CarraChi gel. The CarraChi-ZnSO₄ has higher strain-to-failure (45%) and tensile strength (14.2 MPa) than that of the CarraChi-H₂O (12.1% and 12 MPa), which is ascribed to the cross-linking of Zn²⁺ and the polar groups (-SO₄ and -OH) in the hydrogel network. The bonding effect of CarraChi and Zn²⁺ is also certified by the change of zeta potential in **Fig. 3a**. Moreover, previous works (Adv. Mater. 2019, 31, 1902432; Angew. Chem., Int. Ed. 2016, 55, 15925-15928) reveal that the divalent metal ions can bond with the -SO₄²⁻ and -OH groups in the hydrogel network. Some research work (Environ. Sci. Pollut. Res. 2019, 26, 26254-26264) also uses chitosan and k-carrageenan to absorb different metal ions in polluted water.

Fig. R19. Stress-strain curves of the dry CarraChi gel, CarraChi-H₂O, CarraChi soaked with ZnSO₄, and GF.

Updates in the Revised Manuscript:

We have updated the stress-strain curve in **Fig. 2h** and added the following discussion in the Revised Manuscript:

Line 110-112, Page 6: “Compared to the CarraChi-H₂O, the mechanical strength increased after the Zn²⁺ addition, possibly due to the bonding effect between divalent Zn²⁺ and -SO₄²⁻/-OH functional groups in the hydrogel network.^{40, 41”}

Regarding “visualization proof to demonstrate the gel can prevent dendrite penetration”

To visualize the prevention effect of the CarraChi membrane toward dendrite, we have carried out a deposition experiment using the symmetric cell at 10 mA cm⁻². A smooth Zn deposition is observed with the CarraChi gel electrolyte while the Zn dendrites are mixed with the GF separator, illustrating the effective suppression of Zn dendrites of the CarraChi membrane (Fig. R20). In contrast, the symmetric cell with the GF separator fails when the capacity increases to 20 mAh cm⁻², indicating the short circuit of the symmetric cell. The optical images after cycling in the full cell also reveal that the dendrite penetration occurs with the GF separator (Fig. R21).

Fig. R20. SEM images of deposited Zn. **a**, top and **b**, side views with the CarraChi gel electrolyte (10mA cm^{-2} with deposition capacity of 10mAh cm^{-2}); **c**, top and **d**, side views with the GF separator.

Fig. R21. **a**, Optical images of the pristine and cycled CarraChi gel, GF and Zn anode. SEM images of cycled Zn with **b**, GF separator, and **c**, CarraChi gel electrolyte.

Updates in the Revised Manuscript:

We have also added the following discussion in the Revised Manuscript:

Line 177-179, Page 10: “In contrast, Zn is deposited in the holes of the GF separator with a capacity of 10mAh cm^{-2} (Fig. 3h, i), followed by the short-circuiting of the symmetric cell when the capacity increases to 20mAh cm^{-2} , certifying the separator piercing (Supplementary Fig. 14c, d).”

References

40. Deng Y, et al. Fast Gelation of $\text{Ti}_3\text{C}_2\text{T}_x$ MXene Initiated by Metal Ions. *Adv. Mater.*, **31**, 1902432 (2019).

41. Li D, et al. Double-Helix Structure in Carrageenan-Metal Hydrogels: A General Approach to Porous Metal Sulfides/Carbon Aerogels with Excellent Sodium-Ion Storage. *Angew Chem., Int. Ed. Engl.* **55**, 15925-15928 (2016).

4. Which type of GF is used in this work? The authors should mention the details of GF in the experimental section. This is very important for those who want to repeat this experiment. In addition, the thickness of GF should be at around 50 μm at least. Unfortunately, the thickness of CarraChi gel and GF is different when the authors compare the electrochemical performance, which may affect the conclusions. Please compare the electrochemical performance with the same thickness of CarraChi and GF.

Reply: We thank the reviewer for the helpful comments. GF-A (Whatman) is used in this work, and the details have been added to the **Method**.

The GF-A separator is the thinnest of all GF series with a thickness of 260 μm . However, after being assembled in the batteries, the thickness of the GF-A separator can change apparently to 129 μm . The thickness of the CarraChi membrane also changes to 105 μm after soaking and pressing. Due to the mechanical strength of the CarraChi membrane, CarraChi can be easily made thinner than the thinnest commercial GF-A, which is another advantage of the CarraChi electrolyte.

Thickness is not the major reason affecting the electrochemical performance. Particularly, the CarraChi has strong mechanical properties for preventing the Zn dendrite proliferation, good scalability for large-scale energy storage, and water-bonding properties to prevent water leakage in the open-system, large-format batteries. In contrast, the GF has poor mechanical strength and large-pore structure, easily producing Zn dendrites.

Updates in the Revised Manuscript:

We have also added the GF information in the Revised Manuscript:

Line 338, Page 20: “The pouch symmetric cell was assembled with two identical polished Zn foils ($8 \times 8 \text{ cm}^2$), CarraChi gel, or a GF (type A, Whatman) was used as the separator.”

5. Could the author describe what kind of barrier the gel layer provides for Zn surface diffusion?

Reply: We thank the reviewer for the helpful comments. According to previous reports (Energy Environ. Sci. 2019, 12, 1938-1949; Adv. Mater. 2022, 34, 2202382), the interfacial energy barrier is provided with the gel layer. In our work, the changed zeta potential after ZnSO_4 addition (Fig. 3a in the Revised Manuscript) reveals the effective adsorption of Zn^{2+} by the CarraChi gel, which provides an additional energy barrier for absorbed Zn^{2+} ions to move laterally, as also stated by the reduced 2D diffusion current (Fig. 3c). Therefore, these Zn^{2+} ions are forced to deposit near the sites where the initial adsorption occurred, instead of the fewer sites with low surface energy.

Updates in the Revised Manuscript:

We have also added a related discussion about the deposition barrier in the Revised Manuscript:

Line 167-170, Page 10: “Due to the abundant -OH and $-\text{SO}_4^{2-}$ functional groups, the CarraChi gel adhered on Zn foil would provide an extra interfacial energy barrier to prevent 2D Zn^{2+} diffusion. Meantime, the abundant Zn^{2+} transport channels in the crosslinked gel framework render the Zn^{2+} flux uniform.”

Reference

48. Zhao Z, et al. Long-life and Deeply Rechargeable Aqueous Zn Anodes Enabled by Multifunctional Brightener-Inspired Interphase. Energy Environ. Sci. 12, 1938-1949 (2019).

6. It will be nice if the author could perform some simulations to visualize the electric field at the electrolyte-electrode interface that has been homogenized when gel electrolyte is used in this study.

Reply: We thank the reviewer for the helpful suggestion. Following the suggestion, we carried out finite simulations to visualize the electric field and Zn^{2+} concentration distribution at the electrolyte-electrode interfaces. As shown in Fig. R22, the Zn anode with CarraChi exhibits homogenized Zn^{2+} concentration and electric field distribution along X-axis, which facilitates uniform Zn^{2+} supplement

during the Zn deposition process. In contrast, a higher concentration gradient and nonuniform electric field are observed along X-axis for the Zn anode with GF separator, making it easier for Zn dendrites growth at the Zn/GF interface.

Fig. R22. Simulated Zn-ion concentration distribution inside (a) Zn|CarraChi|Zn and (b) Zn|GFA|Zn symmetric cells; simulated electric potential and electric current distribution inside (c) Zn|CarraChi|Zn and (d) Zn|GF|Zn symmetric cells at 10 mA cm^{-2} (the thin white strips represent GF).

Updates in the Revised Manuscript:

We have provided the Zn^{2+} concentration distribution of Zn|CarraChi|Zn and Zn|GFA|Zn symmetric cells in Fig. 3d and Fig. 3e, respectively, and the electric potential and electric current distribution in Supplementary Fig. 13 (Supplementary Information). We have also added related simulation parameters and discussions in the Revised Manuscript:

Line 171-173, Page 10: “Using finite-element simulation, we further reveal that the uniform Zn deposition is attributed to the homogenized Zn^{2+} concentration and electric field near the Zn-CarraChi interface (Fig. 3d, e and Supplementary Fig. 13).”

Line 361-369, Page 21: “Finite-element simulation of symmetric cells. A two-dimensional continuum model was used to investigate the ionic and potential distributions in Zn metal symmetric cells (Zn|CarraChi|Zn, Zn|GFA|Zn) at a constant current density ($i_0=10 \text{ mA cm}^{-2}$).^{53, 54} In CarraChi and ZnSO₄/GFA electrolytes, there are the mass balance and the charge conservation. The ionic flux density was described by the Nernst-Planck equation with diffusion and migration terms. At the Zn-electrolyte interface, the boundary conditions were set according to the cell test system. The other boundaries were set as no flux. All finite-element simulations were conducted in COMSOL Multiphysics 6.1. The initial conditions and ionic diffusion coefficients were set to be consistent with the electrochemical measurement conditions and results.”

References

53. Lu H, Zhou J, Ye G, Zhou X. Recent Advances in Continuous Models of Electrochemical Supercapacitors. *J. Electrochem.* 24, 517-528 (2018).
54. Su Y, et al. Rational design of a topological polymeric solid electrolyte for high-performance all-solid-state alkali metal batteries. *Nature Commun.* 13, 4181 (2022).

7. “1.26V vs RHE at 5 mA cm^{-2} (S Fig. 12)” Please change Fig. 12 to Fig. 10. In Fig. 10, it indeed verifies HER can not be fully suppressed even using CarraChi gel because the onset potential for HER is almost the same, in which both start around -1.1 V . The difference is that the gel system shows a smaller current density compared with GF. In this case, it is incorrect to claim Fig. 10 demonstrates slower evolution kinetics. Importantly, please use ZnSO₄ electrolyte rather than Na₂SO₄ electrolyte to test LSV.

Reply: We thank the reviewer for pointing out this error. We have corrected the figure number and double-checked all figure numbers.

Regarding “it indeed verifies HER can not be fully suppressed even using CarraChi gel because the onset potential for HER is almost the same”

In the Revised Manuscript, we compared the onset potential instead of the previously selected current density of the LSV. Since the onset overpotential is the applied potential with apparent cathodic currents, we carried out the iR correction for the LSV curves and calculated the onset overpotential

according to previous reports (Nat. Commun. 2019, 10:1348; Sci. Adv. 2015, 1, e1500259). The onset potential is the point where the oblique line extension intersects the line parallel to the horizontal axis. The battery with the CarraChi exhibits a more negative onset potential of -1.40 V, in comparison to that with the GF separator (-1.26 V), illustrating the effective suppression of HER by CarraChi (Fig. R23).

In addition, as we explained above, the HER in ZIBs is not likely to be completely eliminated because metallic Zn is thermodynamically unstable in aqueous environments. In our work, the designed CarraChi gel can mitigate the HER. However, we do not seek to eliminate HER in large-format ZIBs over long-term cycling but use the open system with the gas outlet to avoid H_2 accumulation and battery swelling, thereby circumventing problems caused by HER.

Regarding “please use $ZnSO_4$ electrolyte rather than Na_2SO_4 electrolyte to test LSV.”

The use of Na_2SO_4 instead of $ZnSO_4$ as the supporting electrolyte in HER measurements is a widely accepted method in ZIBs according to the previous reports (Nat. Commun. 2023 14, 641; Adv. Mater. 2022, 34, 2202382; J. Am. Chem. Soc. 2022, 144, 25, 11129–11137; Energy Environ. Sci., 2022, 15, 1638-1646). This is to avoid the deposition of Zn onto the electrode surface during the LSV test if $ZnSO_4$ is used as the supporting electrolyte, which would complicate distinguishing the current contribution of the HER from that of the Zn deposition. The related corrections are updated in the manuscript.

Fig. R23. LSV curves of Zn with/without CarraChi with iR correction.

Updates in the Revised Manuscript:

We have corrected HER curves in the revised Supplementary Information (**Supplementary Fig. 10**).

We have also added the following discussion in the Revised Manuscript:

Line 152-154, Page 9: “The suppressed HER on the Zn anode with CarraChi is also proved by a much lower onset overpotential (−1.40 V vs. standard hydrogen electrode, SHE) than that with a GF separator (−1.26 V vs. SHE) (Supplementary Fig. 10).”

8. The calculation of average CE in the symmetric system in this manuscript is wrong. Please refer to Nat Energy 5, 743–749 (2020). <https://doi.org/10.1038/s41560-020-0674-x> for your average CE calculation.

Reply: We thank the reviewer for commenting on the calculation of average CE. We have calculated the average CE based on the referred paper (Nat. Energy 2020, 5, 743–749). Specifically, a Cu substrate is conditioned by plating/stripping a certain amount of Zn, followed by the deposition of a Zn reservoir (Q_r). A fixed capacity of Zn (Q_c) is cycled 9 times before all the removable Zn (Q_s) is stripped. The calculated average CE is 98% from the voltage-time curves based on this method (Fig. R24).

Fig. R24. Voltage-time curves of the Zn/Cu battery at 1 mA cm^{-2} with the CarraChi gel electrolyte.

However, very few papers reported average CE by this method. To fairly compare the CE with previous reports, we have chosen the more widely-used testing method and used the CE data after stabilizing.

9. It should be more subjective when the authors conclude. The good capacity retention of Zn|CarraChi|ZVO is not only related to the effective suppression of Zn dendrite growth but also the inhibition of self-corrosion and side reactions of Zn foil.

Reply: We thank the reviewer for the helpful comments. We agree that although good capacity retention is mainly ascribed to the prevention of the Zn dendrites, the inhibition of self-corrosion and side reactions also plays an important role in determining good cycling stability.

Following the reviewer's suggestion, we carried out a soaking experiment by leaving the assembled symmetric cells with or without CarraChi for 1 week to research the degree of Zn corrosion. The experimental results show that the Zn in the Zn|CarraChi|Zn cell has a much weaker peak intensity of by-products in the XRD patterns (Fig. R25a). SEM images also reveal that a smooth surface of the Zn foil can be observed with CarraChi (Fig. R25b). In contrast, a Zn foil soaked in 2M ZnSO₄ solution is covered with dendritic by-products of zinc hydroxide sulfate (Fig. R25c). Therefore, the CarraChi gel has a positive effect on mitigating the zinc hydroxide sulfate byproducts.

In all, the good capacity retention of Zn|CarraChi|ZVO is ascribed to the effective suppression of Zn dendrite growth, the inhibition of self-corrosion and side reactions of Zn foil

Fig. R25. a, XRD patterns of Zn foil in contact with ZnSO₄ liquid electrolyte and CarraChi gel

electrolyte. SEM images of the Zn foil after standing for 1 week: **b**, with the CarraChi, and **c**, with ZnSO₄ liquid electrolyte.

Updates in the Revised Manuscript:

We have added the XRD patterns and SEM images of Zn foil soaked with CarraChi gel electrolyte and ZnSO₄ in the revised Supplementary Information (Supplementary Fig. 12). We have also added the following discussion in the Revised Manuscript:

Line 156-160, Page 9: “The CarraChi gel electrolyte also reduces corrosion reactions between the Zn foil and ZnSO₄, which is proved by the weaker peak intensity of by-products in the X-ray diffraction (XRD) patterns and the smooth surface in the SEM images in the anti-corrosion experiment compared to the Zn anode with liquid electrolyte and GF separator (Supplementary Fig. 12).”

Line 274-276, Page 16: “We ascribe the improved cycling stability to the open, refillable battery configuration, as well as the high strength of the CarraChi, inhibition of side reactions, and dendrite-free Zn plating/stripping caused by the uniform ion flux at the Zn-CarraChi interfaces.”

10. the author should explain why ZVO in GF almost shows the same specific capacity as CarraChi before 40 cycles in Figure 5c, while the specific capacity of GF is higher or lower than CarraChi in Figure 5b. Please mention which current density has been used in Figure 5f.

Reply: We thank the reviewer for the valuable comments. In **Fig. 5b**, we show that the full cell with CarraChi has a better rate performance than that with GF. The fluctuating capacity of GF-based batteries may be due to unsatisfying battery assembling or unstable performance using the GF separator. We have conducted more tests of the rate performance of Zn|GF|ZVO (Fig. R26). In particular, a significant capacity decay can be observed for Zn|GF|ZVO, which can be ascribed to the fast dissolution of ZVO in the 2M ZnSO₄.

In **Fig. 5c**, the reason why both cells show the same capacity retention ratio before 30 cycles are that the application of the CarraChi gel electrolyte will not affect the initial capacity. However, CarraChi suppresses the dissolution of the ZVO cathode into the electrolyte during long cycling. We found a

yellow substance adsorbed on the GF separator which is ascribed to the dissolved ZVO (Fig. R27). In contrast, there is no dissolved ZVO on the CarraChi gel. The dissolution of the ZVO cathode into the liquid electrolyte with GF caused significant capacity decay with GF after 30 cycles.

Fig. R26. Comparison of the rate performance of Zn|CarraChi|ZVO and Zn|GF|ZVO batteries at different current densities from 0.2 to 10 A g⁻¹.

Fig. R27. Optical images of CarraChi and GF after standing with ZVO cathode for 1h (the dissolved ZVO adsorbed on GF and there is no dissolved ZVO on CarraChi).

In addition, in **Fig. 5f**, the applied current density is 0.2 A g⁻¹. We have added the testing condition in Fig. 5f.

Updates in the Revised Manuscript:

We have updated the rate performance of Zn|GF|ZVO in **Fig. 5b** and the applied current density 0.2 A g^{-1} in **Fig. 5f** in the Revised Manuscript.

REVIEWER COMMENTS

Reviewer #1 (Remarks to the Author):

The revised version is much improved. The authors adequately respond to the comments raised by the reviewers. The revised version is acceptable for publication.

Reviewer #2 (Remarks to the Author):

I think further revision is required because there are some unclear statements in the authors' rebuttal letter.

1. Considering that the concentration of proton is very low in mild acidic electrolyte conditions (pH above 4.5), I believe the direct reduction of water is the main problem of HER. Direct water reduction can be accelerated by inducing a stronger interaction between water molecules and other solvents or ions than the interaction between the water molecules. This is because, with a stronger interaction, water tends to undergo stronger polarization. Considering the authors' claim that the -OH of CarraChi and H₂O can form a strong bond and CarraChi can interrupt the water network, water and CarraChi should have strong interaction. In this regard, I think water molecules bonding with CarraChi should also be strongly polarized, resulting in a higher degree of direct water reduction for the water molecules bonding with CarraChi near the Zn metal surface. Also, I think the Cathodic challenge should be also considered, too.

2. I think Figure R5 should be provided again without IR correction. The effect of IR is obviously included in real cell operation.

3. I believe the report from Archer's group suggests that the electrolyte's shear modulus is meaningless when the electrolyte has a sufficiently small pore structure. I believe this is not the case for the CarraChi electrolyte.

4. About the CE, it is questionable how the GF cell can have higher CE at certain cycles even though the GF cell suffers from a more severe HER problem than the CarraChi cell.

5. In Zn/Zn symmetric cell, the overpotential for the CarraChi cell is much higher than the GF cell. However, the rate capability of the full cell using CarraChi seems to be better than the GF cell. To make this reasonable, the authors need to show that the CarraChi can decrease the overpotential of the positive electrode reaction. If the high transport kinetics of Zn^{2+} is a really important factor, then why it does not affect the Zn/Zn symmetric cell performance?

Reviewer #3 (Remarks to the Author):

The authors have made good improvement in the revised manuscript. All the questions pointed by the reviewers have been well addressed. I agree the acceptance of the current version of the manuscript for publication.

Point-by-point response to the comments

Reviewer #1: The revised version is much improved. The authors adequately respond to the comments raised by the reviewers. The revised version is acceptable for publication.

Reply: We thank the reviewer for the positive comments and recommendation for publication in Nature Communications.

Reviewer #2: I think further revision is required because there are some unclear statements in the authors' rebuttal letter.

Reply: We sincerely thank the reviewer for providing constructive comments in the two-round reviewing processes, which have improved our work to be more rigorous. To ensure clearer expression, we have supplemented relevant discussions and explanations. Please see our point-by-point responses below.

1. Considering that the concentration of proton is very low in mild acidic electrolyte conditions (pH above 4.5), I believe the direct reduction of water is the main problem of HER. Direct water reduction can be accelerated by inducing a stronger interaction between water molecules and other solvents or ions than the interaction between the water molecules. This is because, with a stronger interaction, water tends to undergo stronger polarization. Considering the authors' claim that the -OH of CarraChi and H₂O can form a strong bond and CarraChi can interrupt the water network, water and CarraChi should have strong interaction. In this regard, I think water molecules bonding with CarraChi should also be strongly polarized, resulting in a higher degree of direct water reduction for the water molecules bonding with CarraChi near the Zn metal surface. Also, I think the Cathodic challenge should be also considered, too.

Reply: We thank the reviewer for the helpful comments.

According to the previous reports (Chem. Soc. Rev., 2020, 49, 3072-3106; Chem. Rev.,

2021, 121, 6654-6695; Adv. Energy Mater. 2020, 2003065), H⁺ reduction predominates even in mild acidic electrolytes and H₂O reduction is dominant in neutral or alkaline electrolytes. Given that the pH of the 2M ZnSO₄ electrolyte is 4.3, we posit that reduction processes of both H₂O and H⁺ may be occurring simultaneously during the electrolysis, which brings challenges in reaching a conclusive determination. In our experiment, a severe corrosion reaction accompanied by HER can be observed during the soaking experiment (Supplementary Fig. 12 and Supplementary Fig. 10), which is different from the slow passivation of Zn foil in pure water, which may indicate that the H⁺ reduction in 2M ZnSO₄ electrolyte is the main cause of HER. Notably, this research primarily emphasizes an application-oriented Zn battery, and we anticipate future collaborations with fellow researchers to undertake in-depth investigations into fundamental aspects of HER.

We agree with the reviewer that water molecules bonding with CarraChi should also be strongly polarized, resulting in a higher degree of direct water reduction. However, the solvated H₂O near the Zn metal is also responsible for the HER during the electrochemical process (Nature, 2021, 600, 81-85; Adv. Mater. 2022, 2206754). The CarraChi can not only bond with free water but also modulate the solvation structures of Zn²⁺, which is certified by the Raman results and zeta potential (Fig. 3a and Supplementary Fig. 9). Thus, the bonding between CarraChi and water molecules does not aggravate the HER. Instead, as the H-bonding network is disrupted by CarraChi, HER is suppressed, as demonstrated by the much lower overpotential (Supplementary Fig. 10).

Regarding the challenge of cathodes, as our open-system battery enables releasing any generated gas in large-format cells, we achieved full cells with much-improved cycling stability in large-format pouch cells. We have compared the performance of Zn|CarraChi|ZVO pouch cells with previously reported ZIB pouch cells in Supplementary Table 7. The Zn|CarraChi|ZVO pouch cell with our proposed battery design exhibits superior cycling stability with high capacity.

2. I think Figure R5 should be provided again without IR correction. The effect of IR is obviously included in real cell operation.

Reply: We thank the reviewer for the helpful comments. Following the reviewer's comments, we have updated the LSV curves without IR correction in the revised Supplementary Information (Supplementary Fig. 10). The onset potentials of the LSV curves are only slightly different with or without IR correction (Fig. R1). In addition, the Zn with CarraChi shows more negative HER onset potential than that with GF, suggesting that the CarraChi can suppress HER.

Fig. R1. LSV curves of Zn with/without CarraChi. **a**, with IR correction and **b**, without IR correction.

Updates in the Revised Manuscript:

We have updated the LSV curves in Supplementary Fig. 10 in the Revised Supplementary Information.

3. I believe the report from Archer's group suggests that the electrolyte's shear modulus is meaningless when the electrolyte has a sufficiently small pore structure. I believe this is not the case for the CarraChi electrolyte.

Reply: We thank the reviewer for the instructive comments. According to Archer's study (J. Am. Chem. Soc. 2014, 136, 7395-7402), mechanical strength is not the only factor that affects dendrites. Small pore structures can also induce dendrite deposition

along the pore, causing short-circuiting. We agree with the reviewer that this conclusion may not be applicable to our gel electrolyte. Based on the Reviewer's comments and literature study, we do not claim the inhibiting effect of mechanical strength on Zn dendrite growth. Still, the gel electrolyte with abundant oxygen-containing groups can induce uniform Zn nucleation and deposition, enabling dendrite-free plating/stripping, which is verified by the SEM images after cycling with the CarraChi gel electrolyte (Fig. 4c).

In the Revised Manuscript, we have revised the discussion on mechanical strength, no longer stating the controversial dendrite-inhibiting capability of mechanical strength.

Updates in the Revised Manuscript:

Line 112-114, Page 6: “Such an improved mechanical property of the CarraChi gel electrolyte is expected to facilitate the electrode fabrication and battery assembling process.”⁴²

4. About the CE, it is questionable how the GF cell can have higher CE at certain cycles even though the GF cell suffers from a more severe HER problem than the CarraChi cell.

Reply: We thank the reviewer for the useful comments. As the GF cells suffer from a more severe HER problem, the cycling performance and CE in the Zn|GF|Cu cells fluctuate significantly from different cells. Therefore, we have retested the CE of more Zn|GF|Cu cells. As shown in Fig. R2, the Zn|CarraChi|Cu cell (Fig. R2a) exhibits higher initial CE, higher average CE, and cycling stability than the Zn|GF|Cu cells (Fig. R2b-f) during the plating/stripping process.

Regarding the CE of the Zn|GF|Cu cells at specific cycles, we do find the CE in certain cycles of some Zn|GF|Cu cells is higher than that of the Zn|CarraChi|Cu cell, mainly found in around 10 cycles. We speculate that this is because of the relatively low initial CE of Zn|GF|Cu cells, leaving unreacted Zn metal on the Cu current collector, which

gradually reacted in subsequent cycles, causing relatively high CE at $\sim 10^{\text{th}}$ cycle in Zn|GF|Cu cells. Nevertheless, the average CE of the Zn|CarraChi|Cu cell is still better than that of Zn|GF|Cu.

Fig. R2. CE curves of Zn plating/stripping in **a**, Zn|CarraChi|Cu and **b-f**, Zn|GF|Cu cells at 5 mA cm^{-2} with a capacity of 1 mAh cm^{-2} .

Updates in the Revised Manuscript:

We have updated the CE curves and data in **Fig. 4d** in the Revised Manuscript.

Fig. 4d. CE curves of Zn plating/stripping in Zn|CarraChi|Cu and Zn|GF|Cu cells.

5. In Zn/Zn symmetric cell, the overpotential for the CarraChi cell is much higher than the GF cell. However, the rate capability of the full cell using CarraChi seems to be better than the GF cell. To make this reasonable, the authors need to show that the

CarraChi can decrease the overpotential of the positive electrode reaction. If the high transport kinetics of Zn^{2+} is a really important factor, then why it does not affect the Zn/Zn symmetric cell performance?

Reply: We thank the reviewer for the helpful comments and suggestions. Indeed, as shown in Fig. R3, the Zn|CarraChi|ZVO cell shows a much lower overpotential than the Zn|GF|ZVO cell from 0.5 to 4 A g^{-1} .

In symmetric cells, the electrolyte/separator determines the ion conductivity and overpotential. In full cells, the cathode largely determines the reaction kinetics due to the complicated Zn^{2+} intercalation/deintercalation mechanisms in the full cells. We thus deduce that in full cells, the transport kinetics of Zn^{2+} in the cathode has a more significant effect on the rate performance than the overpotential caused by the electrolyte/separator.

Thus, as the overpotential in the positive electrode is decreased in the CarraChi cells (Fig. R3), it is reasonable that the rate capability of the full cells using CarraChi is better than the GF cells.

Fig. R3. Voltage profiles of (a) Zn|GF|ZVO and (b) Zn|CarraChi|ZVO at various charge/discharge current densities. (c) Comparison of voltage profiles of Zn|GF|ZVO and Zn|CarraChi|ZVO at 2 A g^{-1} .

Reviewer #3: The authors have made good improvement in the revised manuscript. All the questions pointed by the reviewers have been well addressed. I agree the acceptance of the current version of the manuscript for publication.

Reply: We thank the reviewer for the positive comments and recommendation for publication in Nature Communications.

REVIEWERS' COMMENTS

Reviewer #2 (Remarks to the Author):

I suggest to accept this manuscript.